# Graphite-protected CsPbBr$_3$ perovskite photoanodes functionalised with water oxidation catalyst for oxygen evolution in water

Isabella Poli [1,2], Ulrich Hintermair [1,2], Miriam Regue [2,3], Santosh Kumar [3], Emma V. Sackville[1,2], Jenny Baker [4], Trystan M. Watson [4], Salvador Eslava [2,3] & Petra J. Cameron [1,2]

Metal-halide perovskites have been widely investigated in the photovoltaic sector due to their promising optoelectronic properties and inexpensive fabrication techniques based on solution processing. Here we report the development of inorganic CsPbBr$_3$-based photo-anodes for direct photoelectrochemical oxygen evolution from aqueous electrolytes. We use a commercial thermal graphite sheet and a mesoporous carbon scaffold to encapsulate CsPbBr$_3$ as an inexpensive and efficient protection strategy. We achieve a record stability of 30 h in aqueous electrolyte under constant simulated solar illumination, with currents above 2 mA cm$^{-2}$ at 1.23 V$_{RHE}$. We further demonstrate the versatility of our approach by grafting a molecular Ir-based water oxidation catalyst on the electrolyte-facing surface of the sealing graphite sheet, which cathodically shifts the onset potential of the composite photoanode due to accelerated charge transfer. These results suggest an efficient route to develop stable halide perovskite based electrodes for photoelectrochemical solar fuel generation.

[1] Department of Chemistry, University of Bath, Claverton Down, Bath BA2 7AY, UK. [2] Centre for Sustainable Chemical Technologies, University of Bath, Claverton Down, Bath BA2 7AY, UK. [3] Department of Chemical Engineering, University of Bath, Claverton Down, Bath BA2 7AY, UK. [4] SPECIFIC, Swansea University Bay Campus, Fabian Way, Swansea SA1 8EN, UK. Correspondence and requests for materials should be addressed to U.H. (email: u.hintermair@bath.ac.uk) or to S.E. (email: s.eslava@bath.ac.uk) or to P.J.C. (email: p.j.cameron@bath.ac.uk)

About 1 billion people still lack access to electricity, which has significant implications for healthcare, business and food growing[1]. Stand-alone photovoltaic (PV) systems might help electrify geographical areas with no grid access. However, the intermittent nature of solar energy requires the development of systems to store excess energy to be used whenever the sun is not available. Storage in the form of hydrogen has gained wide interest in recent years[2]. At present, hydrogen production still relies on fossil fuels and is mostly based on methane steam reforming and electrolysis[3]. The electricity used to drive conventional electrolysers can come from PV devices or wind turbines, reducing the reliance on fossil fuels. Photoelectrochemical (PEC) water splitting is an emerging technique that uses semiconductors submerged in aqueous solution to directly split water into oxygen and hydrogen using sunlight. The feasibility of both small and large scale PEC water splitting has been shown to be technologically and economically viable[4]. The development of efficient PEC electrodes can save in construction costs as the PV power generator and electrolyser are combined in just one system[5]. However, the development of efficient visible-light absorber materials that are inexpensive and stable in water is still challenging[6,7].

Halide perovskites are promising materials for solar photovoltaic energy conversion thanks to their low cost, easy processability and high efficiency, with the current world-record efficiency standing at 23.7%[8]. Unfortunately, lead halide perovskites are extremely moisture sensitive and need to be protected from high humidity environments[9,10], severely limiting their use for direct PEC water splitting applications. Several approaches have been used to improve the stability of halide perovskite solar cells towards humidity, such as the use of more hydrophobic alkyl ammonium salts[11,12], surface passivation by hygroscopic polymers, carbon nanotubes and long chain ligands[13–16] or through 2D/3D hybrid structures[17–20]. However, none of these techniques have thus far succeeded in protecting perovskite films when submerged in water for extended periods of time as required for application in direct water splitting.

The first example of sunlight-driven water splitting with halide perovskite tandem cells showed a solar-to-hydrogen (STH) efficiency of 12.3%[21]. Two CH$_3$NH$_3$PbI$_3$ solar cells were connected in series and combined with Ni electrodes coated in a layer of NiFe. The NiFe was used as bifunctional catalyst for both the oxygen and hydrogen reactions, and in the following years other examples of water splitting driven by halide perovskite solar cells have been reported[22–24]. As a result of the inherent moisture sensitivity of most halide perovskite materials, the use of halide perovskites as photoelectrodes in direct contact with aqueous electrolytes remains less explored. The first example of a halide perovskite solar cell used as a photoanode was reported in 2015, where a CH$_3$NH$_3$PbI$_3$ perovskite-based device was protected with a thin Ni layer deposited by magnetron sputtering[25]. Promising initial photocurrents of 12.5 mA cm$^{-2}$ were achieved that gradually dropped to 2.5 mA cm$^{-2}$ within 15 min using S$^{2-}$ as a sacrificial reductant. The same Ni encapsulation technique was later integrated with carbon nanotube/polymer composite protection layers[26] and additional alkyl ammonium salts[27] to extend the operational stability to tens of minutes. A more efficient encapsulation was later demonstrated by Crespo-Quesada et al., who reported a CH$_3$NH$_3$PbI$_3$-based photocathode protected by a low-melting alloy of Bi, In and Sn (Field's metal, FM), achieving stability for about 1.5 h under continuous illumination[28]. The same encapsulation technique was then used to fabricate a CH$_3$NH$_3$PbI$_3$ halide perovskite-based photoanode that lasted up to 6 h[29]. FM was also used to protect a cesium formamidinium methylammonium (CsFAMA) halide perovskite-based photocathode in tandem with a BiVO$_4$ photoanode that showed

stabilities of up to 20 h[30]. Finally, Zhang et al. reported a CH$_3$NH$_3$PbI$_3$-based photocathode encapsulated with Ti foil sputtered with Pt, which operated for 12 h under continuous illumination in water[31]. While some of these results appear promising, all of the approaches reported so far rely on expensive materials or sophisticated fabrication techniques.

Initial results on PEC water splitting with halide perovskite materials used the archetypal CH$_3$NH$_3$PbI$_3$ and CsFAMA triple-cation mixed halide perovskite that produces photovoltages around 1 V, which alone is below 1.23 V required for water splitting. Wider-bandgap materials can provide higher photovoltages in the range required for water splitting. CsPbBr$_3$ for instance, a fully inorganic compound stable up to 500 °C under N$_2$, has an energy bandgap of 2.3 eV, and solar cells with very high open-circuit voltages have been reported[32–40]. For example, we have used CsPbBr$_3$ in mesoporous carbon solar cells to obtain efficiencies of 8.2% and open-circuit voltages up to 1.45 V[41]. Recently, CsPbBr$_3$ was used to fabricate a photocathode for hydrogen generation that lasted more than 1 h in water but it used again the expensive Field's metal encapsulation[42].

Here, we present the first report of efficient and stable light-driven water oxidation using CsPbBr$_3$-based photoanodes in aqueous solution. A 20-μm-thick mesoporous carbon (m-c) layer combined with a commercial hydrophobic graphite sheet (GS) is used to effectively protect the highly moisture-sensitive CsPbBr$_3$ from degradation in water. The TiO$_2$|CsPbBr$_3$|m-c|GS electrodes operate in water across a very wide pH range (2–13), exhibiting photocurrents of up to 3.8 mA cm$^{-2}$ at an applied bias of 1.23 V$_{RHE}$ in acidic solution. Even more impressively they are stable for 30 h of operation with currents above 2 mA cm$^{-2}$ at 1.23 V$_{RHE}$ in aqueous alkaline electrolyte (pH 12.5) under continuous simulated solar illumination of 1 sun (AM 1.5 G). We further show the versatility of our photoanode architecture by functionalising the GS surface with an Ir-based water oxidation catalyst (WOC). The catalyst lowers the onset potential of the photoanode by ~100 mV. The photovoltage measured in aqueous solution is 1.2 V at pH 3.5 and higher than 1.3 V at pH 8.8–12.5. For the first time for a perovskite-based photoanode the Faradaic efficiency of light-driven water oxidation can be assessed, showing our CsPbBr$_3$-based photoanodes to perform photoelectrochemical O$_2$ evolution from water with a record efficiency of 82.3%.

## Results

**CsPbBr$_3$ perovskite device preparation.** CsPbBr$_3$ is a fully inorganic halide perovskite material, inherently unstable in water like most other organic–inorganic halide perovskites. Supplementary Fig. 1a, b shows the X-ray diffraction (XRD) patterns and UV-vis absorption of a CsPbBr$_3$ film before and after it was dipped in water for 1 s. The initial XRD spectrum was in good agreement with the orthorombic perovskite structure[43], however, after immersion in water, its distinct (100), (110) and (200) diffraction peaks rapidly disappeared and peaks characteristic of the precursor materials PbBr$_2$ and CsBr appeared. While pristine CsPbBr$_3$ has a bandgap of 2.35 eV with an absorption onset corresponding to 530 nm giving rise to its yellow colour, after immersion in water only a semitransparent white material remains (PbBr$_2$). Therefore, in order to use CsPbBr$_3$ in aqueous PEC cells for solar water splitting, effective protection strategies need to be developed as for any other halide perovskite.

We prepared our perovskite photoanodes based on the architecture of a standard carbon perovskite solar cell. Generally, perovskite solar cells are made using an absorber material sandwiched between an electron transport layer (ETL) and a hole transport layer (HTL). In these cells, a compact layer of TiO$_2$ deposited by spray pyrolysis and a doctor-bladed mesoporous

carbon layer (m-carbon or m-c) were used as the ETL and HTL, respectively, as shown in Fig. 1a (Supplementary Fig. 2 shows a schematic energy band diagram). CsPbBr$_3$ solar cells with similar carbon-based architectures have already shown good stability over 3 months in humid air, thanks to the hydrophobic nature of the m-carbon HTL[44]. Indeed, the water contact angle (WCA) of the mesoporous carbon layer used here exceeded 90° (Supplementary Fig. 3). Figure 1b shows a cross-sectional scanning electron microscopy (SEM) image of the as-prepared CsPbBr$_3$ structure with a 20-μm-thick m-carbon layer acting as conducting contact. Energy dispersive X-ray (EDX) mapping images of the CsPbBr$_3$ solar cell are shown in Supplementary Fig. 4, confirming the formation of a TiO$_2$|CsPbBr$_3$|m-c stack. Figure 1c shows the FTO|TiO$_2$|CsPbBr$_3$ interfaces with thin layers of TiO$_2$ and CsPbBr$_3$ of about 50 and 350 nm, respectively.

We first prepared photoanodes with a TiO$_2$|CsPbBr$_3$|m-c configuration (schematic illustration in Supplementary Fig. 5a), where the 20 μm-carbon layer aimed to provide waterproof protection to the moisture-sensitive CsPbBr$_3$. Supplementary Fig. 5b shows the current density-voltage (JV) curve of a typical TiO$_2$|CsPbBr$_3$|m-c solar cell measured under AM 1.5 G. and solar simulated light at 100 mW cm$^{-2}$. As shown in Supplementary Fig. 5c and Supplementary Note 1, the TiO$_2$|CsPbBr$_3$|m-c photoanode in pH 9 aqueous electrolyte exhibited a photocurrent density of ~0.4 at 1.23 V$_{RHE}$. Chronoamperometry was performed at pH 9 and 13 at 1.23 V$_{RHE}$ and is shown in Supplementary Fig. 6a and Supplementary Fig. 7. Promisingly, the photoanode retained 70% of its initial value over 20 min testing and H$_2$

bubbles were generated on the Pt counter electrode (Supplementary Fig. 6b). The photocurrent density was also measured with monochromatic light, showing that CsPbBr$_3$ was still absorbing visible light at wavelengths $\lambda$ < 530 nm after 15 min of immersion in water at pH 9 (see Supplementary Fig. 6c). This photocurrent density could not be assigned to the TiO$_2$ layer or to the formation of PbBr$_2$ since a control photoanode consisting of TiO$_2$|PbBr$_2$|m-c generated a maximum of 0.067 mA cm$^{-2}$ at 1.23 V$_{RHE}$ (Supplementary Fig. 8).

**Protection of perovskite device in water.** In order to more efficiently protect CsPbBr$_3$ from water, a 25 μm self-adhesive graphite sheet (GS) was placed on top of the mesoporous carbon layer, to create TiO$_2$|CsPbBr$_3$|m-c|GS photoanodes (see Fig. 2a). While the mesoporous carbon acts as an effective HTL due to its intimate contact with the perovskite absorber layer, the GS offers a more compact seal that can protect the halide perovskite more efficiently from liquid water percolation (see Hg porosimetry of m-c and GS in Supplementary Fig. 9 and Supplementary Note 2 showing differences in porosity). Similar to mesoporous carbon, the GS is distinctly hydrophobic (measured WCA is 98°, shown in Supplementary Fig. 10) and it offers excellent thermal and electrical conductivity (1600 W m$^{-1}$ K$^{-1}$ and 20,000 S cm$^{-1}$, respectively). Figure 2b shows that the JV curves of the TiO$_2$|CsPbBr$_3$|m-c|GS measured as solar cells before and after 2 h of PEC operation in water were unchanged, indicating extraordinary water stability. As shown from the linear sweep voltammetry (LSV) curve in Fig. 2c, the photocurrent steeply rose from

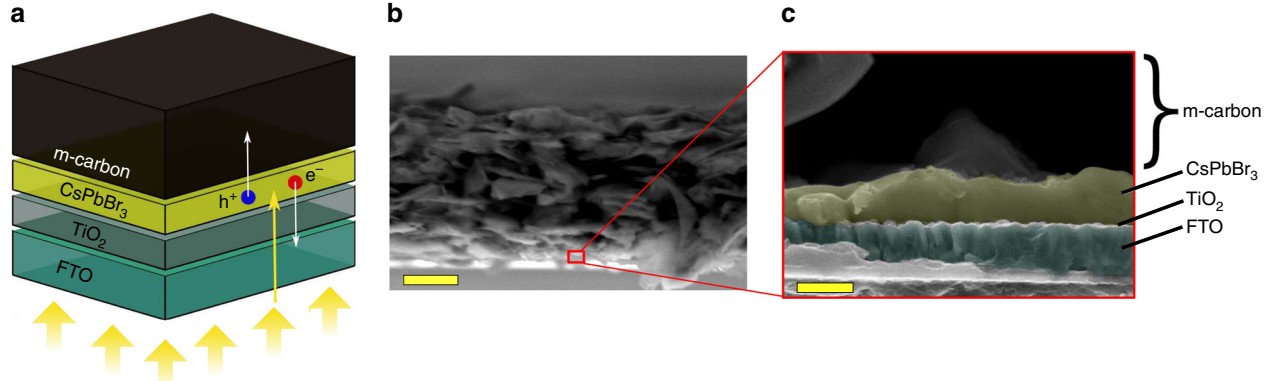

**Fig. 1** Configuration of CsPbBr$_3$ devices. **a** Structure of standard carbon solar cell (illuminated through the FTO glass as indicated by the yellow arrows). Electron-hole (e$^-$/h$^+$) pairs are generated in the CsPbBr$_3$ thin layer, then e$^-$ and h$^+$ are collected in the TiO$_2$ and m-carbon layers, respectively. **b** Cross-sectional scanning electron microscopy (SEM) image of the as-prepared CsPbBr$_3$ perovskite solar cell (scale bar 10 μm). **c** Magnification of the cross-sectional SEM image to show the FTO|TiO$_2$|CsPbBr$_3$ interfaces (scale bar 500 nm)

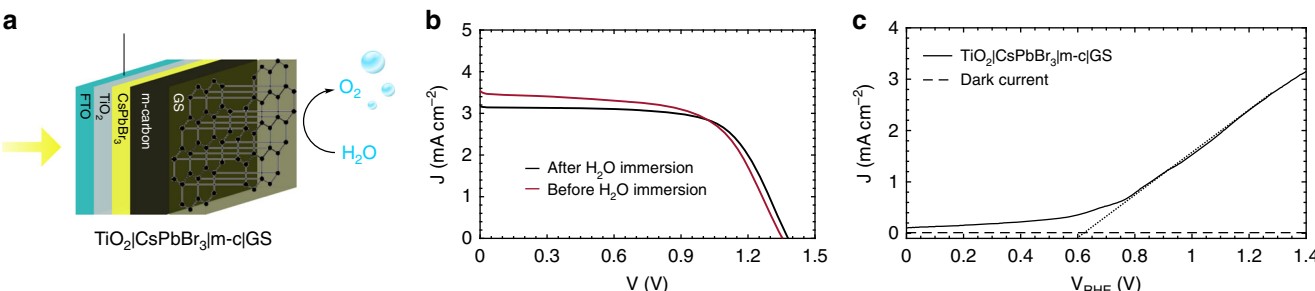

**Fig. 2** TiO$_2$|CsPbBr$_3$|m-c|GS as solar cell and photoanode. **a** Schematic illustration of the CsPbBr$_3$ photoanode for PEC O$_2$ evolution: TiO$_2$|CsPbBr$_3$|m-c|GS uses commercial conductive GS on the m-carbon layer; The photoanode is illuminated through the FTO glass. **b** Photovoltaic current density-voltage curves measured under simulated AM 1.5 G solar light (100 mW cm$^{-2}$) before and after 2 h of immersion in water. Curves shown are measured in reverse scan. **c** Typical linear sweep voltammetry (LSV) measured under PEC conditions at a scan rate of 20 mV s$^{-1}$, measured in 0.1 M KNO$_3$ at pH 4.3, under continuous simulated AM 1.5 G solar illumination, 1 sun

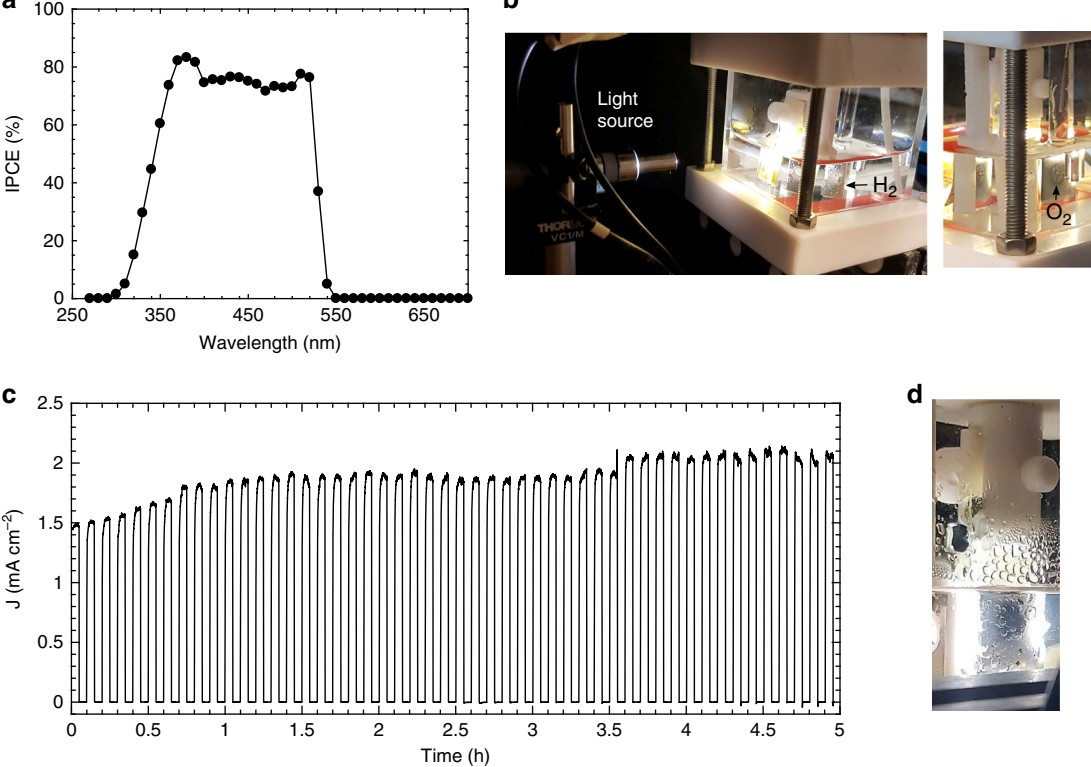

**Fig. 3** TiO$_2$|CsPbBr$_3$|m-c|GS PEC performance in water. **a** Incident photon-to-current efficiency (IPCE) measured in aqueous 0.1 M KNO$_3$ solution with the pH adjusted to 12 with KOH; electrode was subjected to monochromatic light irradiation at 1.23 V$_{RHE}$. **b** Photographs of the PEC cell under operation, showing H$_2$ and O$_2$ gas bubbles evolving from the counter electrode (Pt) and the photoanode, respectively. **c** Chronoamperometric trace recorded at an applied potential of 1.23 V$_{RHE}$ in KOH electrolyte solution at pH 12.5, under chopped simulated solar light irradiation (AM 1.5 G, 100 mW cm$^{-2}$). **d** Photographs of the photoanode immersed in the electrolyte solution after 18 h of continuous operation, showing oxygen bubbles evolving from the active area

0.6 V$_{RHE}$ on, achieving a photocurrent density of 2.5 mA cm$^{-2}$ at 1.23 V$_{RHE}$.

The wavelength dependence of the incident photon-to-current efficiency (IPCE) measured in aqueous solution in Fig. 3a showed an onset wavelength of 530 nm for the TiO$_2$|CsPbBr$_3$|m-c|GS, which corresponds exactly to the onset light absorption of CsPbBr$_3$ in Supplementary Fig. 1b. An IPCE efficiency of about 70% was measured at 500 nm. In contrast, an IPCE efficiency lower than 10% was measured at 500 nm for the TiO$_2$|CsPbBr$_3$|m-c in water (Supplementary Fig. 11), suggesting that the CsPbBr$_3$ partially degraded when protected by the m-c layer on its own. The GS, however, proved to be a particularly effective encapsulation layer which did not hinder charge transport. This was confirmed by the JV curves of TiO$_2$|CsPbBr$_3$|m-c measured before and after applying the GS on the surface, which are shown in Supplementary Fig. 12 and Supplementary Table 1. The V$_{oc}$, J$_{sc}$ and FF of the device decreased by only about 5% after the GS was applied on the surface, confirming that holes could be properly extracted through the GS layer. Photographs in Fig. 3b show O$_2$ and H$_2$ gas bubbles evolving from the GS active area of the CsPbBr$_3$ based photoanode and the Pt counter electrode, respectively.

Figure 3c shows chronoamperometry recorded at an applied potential of 1.23 V$_{RHE}$ in electrolyte at pH 12.5. The TiO$_2$|CsPbBr$_3$|m-c|GS photoanodes were stable under chopped illumination in basic aqueous electrolyte for more than 5 h and the photoanode's current increased from 1.5 to 2 mA cm$^{-2}$ during this time. This represents the first example of halide perovskite-based photoeletrodes, protected with inexpensive

protection layers, which can demonstrate stable photocurrents for hours.

To investigate the long-term stability of the CsPbBr$_3$-based photoanode, chronoamperometry was recorded at a constant applied potential of 1.23 V$_{RHE}$ in aqueous solution (0.1 M KNO$_3$) under continuous illumination. The evolution of the photocurrent density with time at different pH until the end of life is shown in Supplementary Fig. 13. The device achieved a lifetime of 34 h in an alkaline electrolyte and 23 h in near-neutral solution, while the longest lifetime tested in acidic solutions was 7.8 h. Figure 3d shows a photograph of the PEC cell after 18 h of operation, with oxygen bubbles still emerging from the active area of the photoanode inside the PEC cell. The end of life of all devices tested at different pH was always caused by delamination and fracture of the GS surface (Supplementary Fig. 14). In contrast to previous reports in the literature, which all reported constant decreasing photocurrent under continuous illumination[29,31], we observed an increasing photocurrent during the measurement, especially in near-neutral pH, where the photoanode's current density gradually increased under illumination reaching a peak of 2.5 mA cm$^{-2}$ after 16.7 h (Supplementary Fig. 13). We believe that soaking effects are responsible for causing the increase in photocurrents. Tensile strength measurements showed that the GS material used to encapsulate the moisture-sensitive perovskite layer does experience changes in its mechanical properties when exposed to aqueous electrolyte at pH 7 (as shown in Supplementary Fig. 15). The decrease in stiffness observed likely reflects swelling during soaking that can expose a higher surface area to the electrolyte, thereby increasing water

oxidation performance. To further verify this effect of soaking, we tested a TiO$_2$|CsPbBr$_3$|m-c|GS photoanode with intermittent electrolyte soaking in the dark at open circuit. The results in Supplementary Fig. 16 show that the photocurrent density increased every time the photoanode was illuminated again following a period of soaking. Another possible explanation for the increase in photocurrent is possible structural modification of the CsPbBr$_3$ due to halide migration under light soaking[45]. Mosconi et al. proposed that Frenkel defects ($V_x^+/X_i^-$) in halide perovskites heal during irradiation due to lower energy barriers for halide (X) migration in the photoexcited state[46]. Indeed, we previously showed that the efficiency of CsPbBr$_3$ carbon solar cells increased under light soaking[41]. Gradual increase in current over time has been also recently reported for two dimensional/three dimensional perovskite carbon solar cells and triple-cation perovskite devices[47].

In order to further optimise the photoanode and enhance the stability of the GS when exposed to aqueous electrolyte, we measured the performance of TiO$_2$|CsPbBr$_3$|m-c|GS using a 70-μm-thick GS (GS70). Even though a thicker GS was used, charges were still efficiently extracted when testing the device as a solar cell before PEC testing (Supplementary Fig. 17 and Supplementary Table 2). Figure 4 shows chronoamperometry of TiO$_2$|CsPbBr$_3$|m-c|GS70 photoanodes recorded at a constant applied potential of 1.23 V$_{RHE}$ in aqueous solution (0.1 M KNO$_3$, pH 7) under continuous illumination. The photoanode exhibited photocurrents above 2 mA cm$^{-2}$ for about 30 h. Eventually, the silicone-epoxy resin sealing on the slide edges partly degraded in the electrolyte solution, letting some water pass through it, which slowly degraded the CsPbBr$_3$ material to PbBr$_2$ (Supplementary Fig. 18). The GS70 was still intact at the end of the experiment and as mentioned above the failure was due to degradation of the sealant. In contrast, failure in devices with GS25 protecting layers occurred due to fracture of the GS after extended operation in water. This clearly indicates that thicker GS protection layers, along with better sealing materials, can extend the lifetime of these composite photoanodes even further.

**Functionalisation of GS with a water oxidation catalyst**. To test the versatility of our encapsulated perovskite-based photoanodes and to further improve their efficiency at lower potentials, the addition of an iridium-based water oxidation catalyst (WOC)[48–50] was explored. The [Ir(μO)(pyalk)(H$_2$O)$_2$]$_2^{2+}$ catalyst chosen has previously been shown to rapidly and irreversibly adsorb to the surface of indium tin oxide (ITO) and α-Fe$_2$O$_3$ as a

minimally thin molecular monolayer[50–53]. Gratifyingly, we also found it to robustly bind to graphitic surfaces after simply floating the GS on an activated WOC solution at room temperature for 16 h (Fig. 5a). Supplementary Fig. 19 shows EDX mapping images and the elemental composition of the GS|WOC, providing direct evidence for the presence of the Ir-WOC on the surface and showing its localisation around oxidic edges on the GS surface[50]. Evidence for the presence of the catalyst was also provided by electrochemical measurements of the GS|WOC sheet. Figure 5b shows cyclic voltammetry (CV) with a clear catalytic wave for water oxidation starting at about 1.4 V$_{RHE}$[54]. As previously reported for the Ir-WOC in solution[49] and on ITO surfaces[50], the quasi-reversible redox process centred around 1 V$_{RHE}$ appears in acidic media and corresponds to the Ir$^{III}$/Ir$^{IV}$ redox couple[50]. Comparing CVs of both bare and functionalised GS across a pH range of 2–13 (Supplementary Fig. 20) showed the overpotentials for oxygen evolution to increase with increasing pH for both materials, but with distinctly lower overpotentials in the presence of the Ir-WOC at all pH values (Fig. 5c).

These pre-functionalised GS|WOC sheets could easily be used to fabricate TiO$_2$|CsPbBr$_3$|m-c|GS|WOC photoanodes as illustrated in Fig. 6a using a straightforward procedure under ambient conditions. When measured as a photoanode in aqueous solution at pH 2.5, a cathodic shift in the onset potential $V_{on}$ of 100 mV was observed with respect to unfunctionalised TiO$_2$|CsPbBr$_3$|m-c|GS (Fig. 6b). Moreover, the PEC photocurrent density was found to rise faster with applied potential compared with TiO$_2$|CsPbBr$_3$|m-c|GS, a clear sign of improved hole transport and O$_2$ evolution kinetics as seen previously for the same Ir-WOC on haematite photoanodes[52]. The shift in the $V_{on}$ to more negative potentials was observed also at higher pHs 4 and 7 (Supplementary Fig. 21 and Supplementary Note 3) and it was in agreement with Tafel plots (Supplementary Fig. 22). Supplementary Fig. 23 shows CVs for complete TiO$_2$|CsPbBr$_3$|m-c|GS|WOC photoanodes at pH 2.5, 4 and 7. The photocurrent density was higher and the onset potential shifted to more negative values at low pH, as already evidenced by the overpotential trend observed in Fig. 5c as a function of pH.

Open-circuit measurements were performed in the dark and under illumination in different electrolytes (Fig. 7a). The difference between the dark open-circuit voltage and that under illumination determines the photovoltage $\Delta V_{ph}$ of the photoanode[52]. TiO$_2$|CsPbBr$_3$|m-c|GS and TiO$_2$|CsPbBr$_3$|m-c|GS|WOC showed similar open-circuit potential (OCP) and $\Delta V_{ph}$ at equilibrium, showing again that the enhanced activity of TiO$_2$|CsPbBr$_3$|m-c|GS|WOC is not a thermodynamic effect but the result of enhanced charge transfer kinetics. When 1.23 V$_{RHE}$ were applied in a three-electrode PEC system, similar water oxidation photocurrents for photoanodes with and without WOC were obtained (Supplementary Figs 24 and 25 and Supplementary Note 4). The long-term performance and stability of TiO$_2$|CsPbBr$_3$|m-c|GS|WOC showed the addition of the WOC to afford almost 20% higher initial photocurrents of 3.5 mA cm$^{-2}$ and nearly three times longer lifetime of 17 h in acidic electrolyte than the bare TiO$_2$|CsPbBr$_3$|m-c|GS under the same conditions (Fig. 7b).

**O$_2$ production with functionalised perovskite photoanode**. The amount of O$_2$ produced over time by a TiO$_2$|CsPbBr$_3$|m-c|GS|WOC was enough to be monitored using an oxygen probe in the headspace of a gas-tight PEC cell. Figure 8a shows the detected and predicted O$_2$ evolution as a function of time under continuous simulated solar light illumination (1 sun, AM 1.5 G) at an applied voltage of 1.23 V$_{RHE}$, with the predicted amount of O$_2$ calculated from the photocurrent generated as shown in

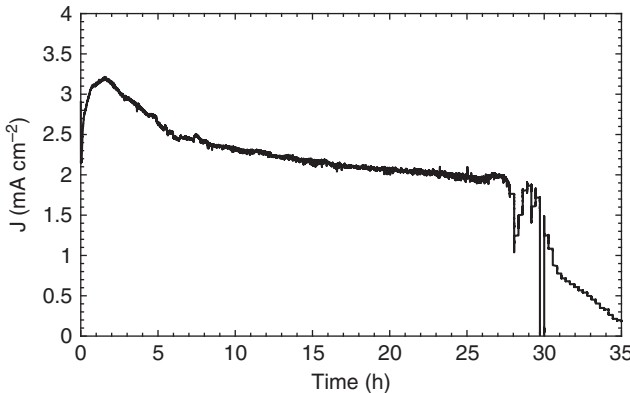

**Fig. 4** Long-term stability of TiO$_2$|CsPbBr$_3$|m-c|GS70 in water. Chronoamperometric trace of TiO$_2$|CsPbBr$_3$|m-c|GS70 recorded at an applied potential of 1.23 V$_{RHE}$. 0.1 M KNO$_3$ electrolyte solution pH 7, under continuous simulated solar light irradiation (AM 1.5 G, 100 mW cm$^{-2}$)

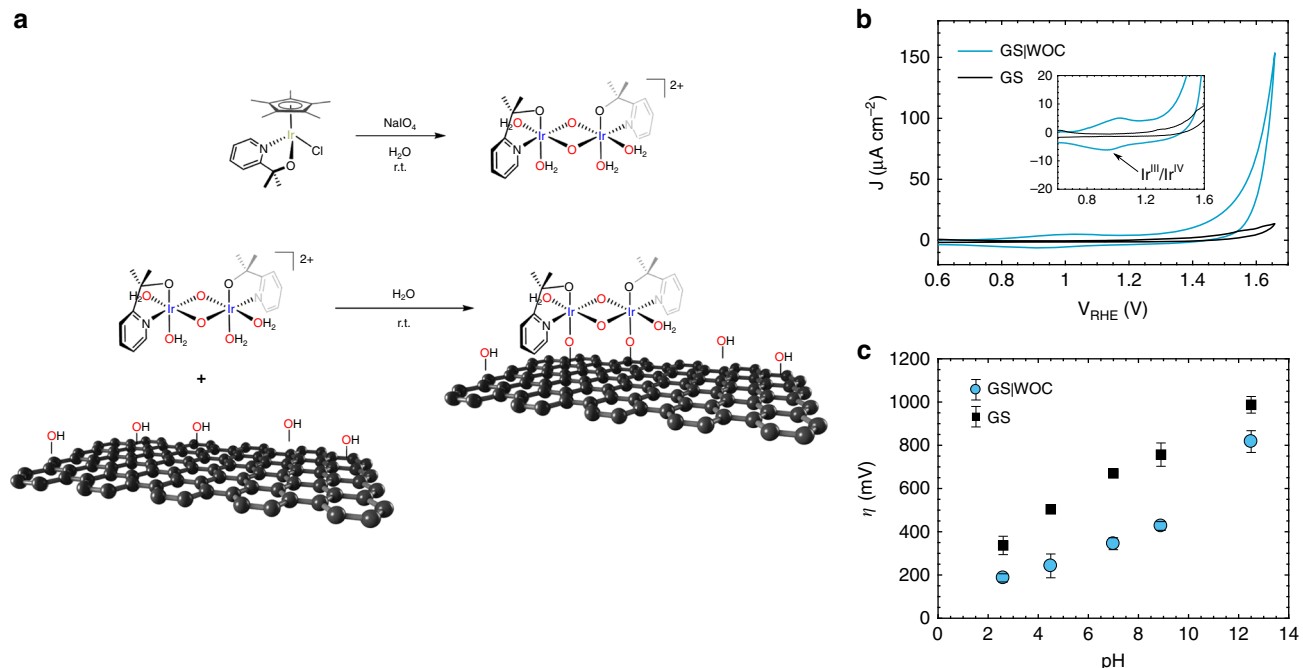

**Fig. 5** Formation of the WOC and functionalisation of the GS surface. **a** Oxidative activation of the iridium precursor (top) and functionalisation of the GS surface (bottom). **b** Cyclic voltammetry (CV) scan of GS and GS functionalised with the Ir-WOC measured in 0.1 M $KNO_3$ at pH 2.6, with scan rate of 50 mV s$^{-1}$. The inset shows a close-up of the potential axis, where the $Ir^{III}/Ir^{IV}$ couple can be seen. **c** Electrode overpotentials at 10 μA cm$^{-2}$ as a function of the pH of the solution. The error bars represent standard deviation

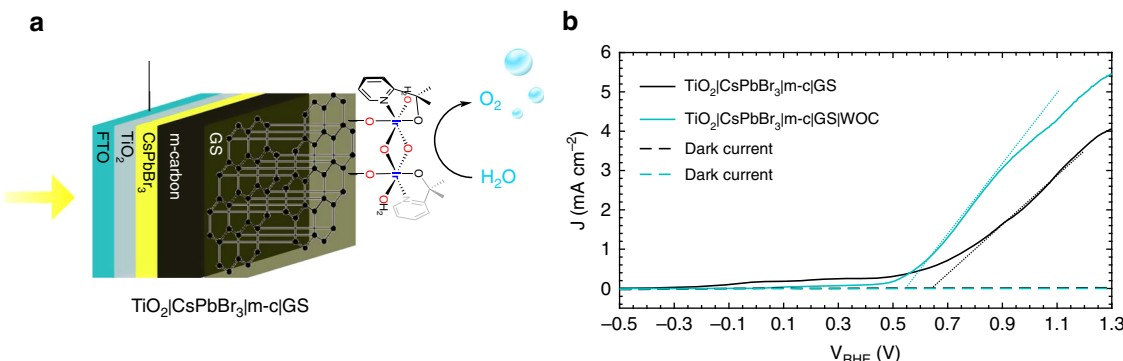

**Fig. 6** $TiO_2|CsPbBr_3|$m-c$|GS|WOC$ as solar cell and photoanode. **a** Schematic illustration of the $CsPbBr_3$ photoanode for PEC $O_2$ evolution: $TiO_2|CsPbBr_3|$m-c$|GS|WOC$ uses commercial conductive GS functionalised with an Ir-based catalyst. The photoanode is illuminated from the back side. **b** Linear sweep voltammetry (LSV) of a $TiO_2|CsPbBr_3|$m-c$|GS$ and a $TiO_2|CsPbBr_3|$m-c$|GS|WOC$ measured at a scan rate of 20 mV s$^{-1}$, in a 0.1 M $KNO_3$ electrolyte solution with pH 2.5 (pH adjusted with $H_2SO_4$)

Supplementary Fig. 26 and Supplementary Note 5. Figure 8b shows the Faradaic efficiency of the light-driven water oxidation, which reached values as high as 82.3% using this unoptimised small-scale laboratory setup (Supplementary Discussion). This value compares favourably with other photoanode materials such as $WO_3$ and $TiO_2$[55,56], and represents the first successful measurement of a Faradaic efficiency of a stable perovskite-based photoanode in aqueous solution.

The photovoltages of about 1.35 V measured in neutral to alkaline media (pH = 8.8 and 12.5 V) exceeded the thermodynamic 1.23 V required for electrolysis of water. In practice, most PEC devices require photovoltages in the range of 1.7–2 V to effectively overcome the kinetic limitations of proton reduction and water oxidation at the electrolyte interface and drive unbiased solar water splitting[57,58]. Indeed, when an unbiased two-electrode measurement was performed with $TiO_2|CsPbBr_3|$m-c$|GS$ and Pt

electrodes at pH 2.5, very low photocurrent of 0.05 mA cm$^{-2}$ was measured (Fig. 9a). Excitingly, when a surface functionalised $TiO_2|CsPbBr_3|$m-c$|GS|WOC$ was tested under the same conditions, a photocurrent of 0.1 mA cm$^{-2}$ was sustained for more than 5 min (Fig. 9b).

Unfortunately the corresponding amount of $O_2$ produced was below the detection limit of our oxygen probe (theoretically we generated 28 nmol after 2 h in de-aerated solution, Supplementary Fig. 27), but the photocurrents observed for $TiO_2|CsPbBr_3|$m-c$|GS|WOC$ constitute an exciting glimpse at the possibility of unbiased solar water splitting that might become possible with these photoanode architectures after some further optimisation.

In conclusion, we have demonstrated a straightforward versatile and highly effective approach for the fabrication of stable inorganic halide perovskite-based photoanodes using an easily fabricated mesoporous carbon layer and commercially

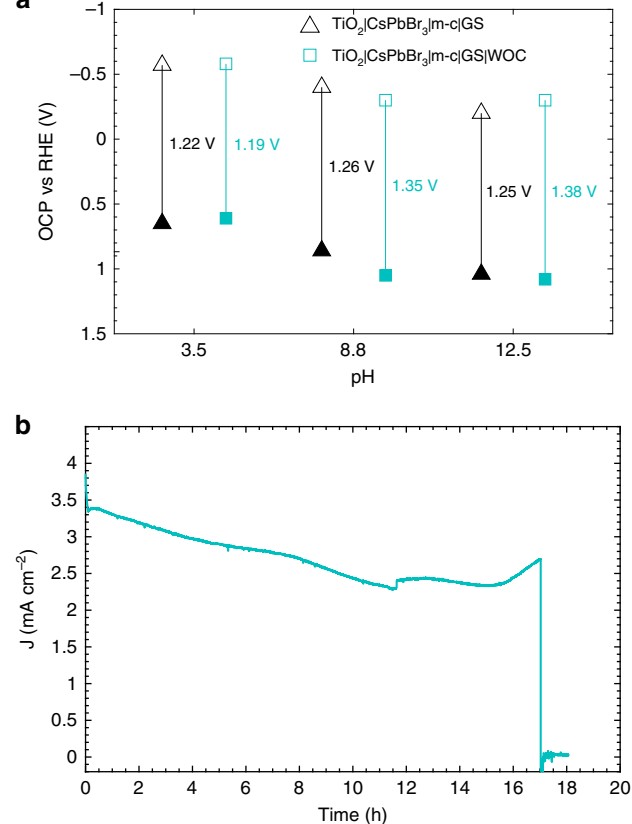

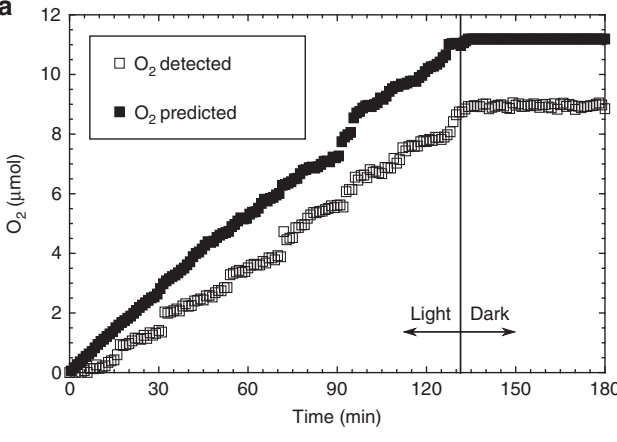

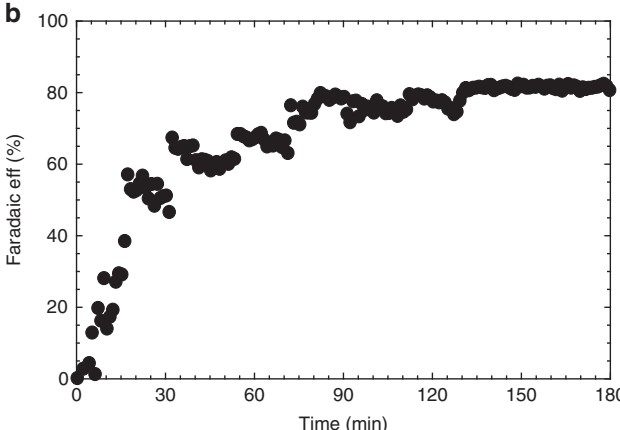

**Fig. 7** Comparison of $TiO_2|CsPbBr_3|m-c|GS$ with and without Ir-WOC. **a** Open-circuit potential (OCP) of photoanodes measured in 0.1 M $KNO_3$ solutions with different pH; Measurements performed in the dark are shown with solid symbols, while the ones performed under illumination are shown with open symbols; The photovoltage $\Delta V_{ph}$ measured for each sample is given. **b** Chronoamperometric trace of $TiO_2|CsPbBr_3|m-c|GS|$ WOC recorded at an applied potential of 1.23 $V_{RHE}$. The electrolyte was 0.1 M $KNO_3$ adjusted to pH 2.5 with $H_2SO_4$, under continuous simulated solar light irradiation (AM 1.5 G, 100 mW cm$^{-2}$)

**Fig. 8** $O_2$ evolution performance on $TiO_2|CsPbBr_3|m-c|GS|WOC$. **a** Detected (open square) and predicted (close square) $O_2$ production under continuous simulated solar light irradiation (AM 1.5 G, 100 mW cm$^{-2}$), in 0.1 M $KNO_3$ adjusted to pH 3.5 with $H_2SO_4$. **b** Faradaic efficiency for $O_2$ evolution

available graphite sheet as hole transport and protection layers. This encapsulation technique was effective in increasing the photocurrent density of the $CsPbBr_3$-based photoanodes from 0.4 to more than 2 mA cm$^{-2}$ at 1.23 $V_{RHE}$, and at the same time protected the perovskite from degradation by the aqueous electrolyte. The so-sealed photoanodes worked efficiently in aqueous electrolytes with IPCE values above 70% for direct light-driven water oxidation in aqueous solution, showing good activity over a wide pH range of 2–13. Photocurrents above 2 mA cm$^{-2}$ were obtained for 30 h of continuous illumination in alkaline aqueous solution. We have also demonstrated the versatility of our photoanode device by effectively functionalising the electrolyte-facing surface of the GS with an Ir-based WOC to improve the onset potential of the photoanode by 100 mV in acidic solutions via improved charge transport kinetics. High Faradaic efficiencies for $O_2$ evolution of up to 82.3% were achieved over 2 h of continuous simulated sunlight irradiation with such a $TiO_2|CsPbBr_3|m-c|GS|WOC$ photoanode.

These composite cells are remarkably easy to synthesise without the need for high temperature or vacuum techniques. A working device could be fabricated by a single person using standard laboratory techniques and inexpensive materials and methods. The GS protection strategy used here for oxygen evolution in water can also be used to enhance the lifetime of perovskite solar cells, where the flexible nature of graphite may allow for roll-to-roll processing for efficient large scale manufacturing. We believe that this design represents a promising lead for using inexpensive, high performance but inherently moisture and water-sensitive perovskite materials in integrated photoelectrochemical cells for solar energy conversion. The flexibility of the underlying perovskite material and option of adding matched oxygen-evolution catalysts allows for a high degree of variability that promises application in a wide range of different architectures.

## Methods

**Photoanode fabrication**. FTO glass TEC 7 (Sigma-Aldrich) was etched with zinc powder and HCl (2 M). It was then cleaned in 2 vol% Hellmanex solution in water, followed by de-ionised (DI) water, acetone (VWR), 2-propanol (IPA, VWR) and ethanol (EtOH,VWR) before being treated with UV-Ozone cleaning (in ProCleaner PLUS for 20 min). A compact $TiO_2$ layer was deposited by spray pyrolysis, using a hand-held atomiser to spray 0.2 M solution of titanium diisopropoxide bis (acetylacetonate) (75 wt%, Sigma-Aldrich) in EtOH onto the substrates held at 500 °C. Substrates were then sintered at this temperature for 10 min. $PbBr_2$ solution (1 M in DMF) was spin coated on top of the substrates held at 70 °C at 2500 rpm for 30 s and each cell was further sintered at 70 °C for 30 min. The cell was then immersed in the CsBr solution (17 mg ml$^{-1}$ in methanol) kept at 50 °C in a vertical staining jar for 30 min before being annealed at 150 °C for 30 min. A carbon paste (black carbon + graphite, Gwent Electronic Materials) was doctor bladed as top contact on top of the $CsPbBr_3$ film. Devices were further post-annealed at 380 °C for 30 min in air for the mesoporous carbon (m-c) layer formation. A graphite thermal sheet (RS, Panasonic, 1600 W m$^{-1}$ K$^{-1}$, 180 × 115 mm, 0.025 mm, self-adhesive on one side) was stuck by hand onto the mesoporous carbon top layer. The device was finally sealed (except an active area of 0.25 cm$^2$) with commercial silicone and epoxy resin left to harden out at room temperature overnight.

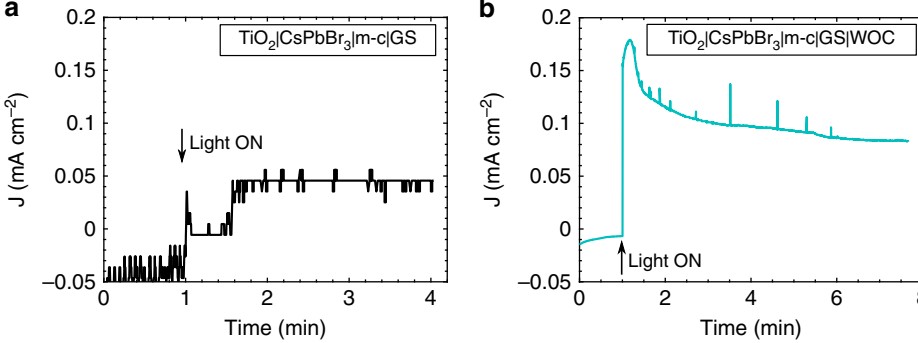

**Fig. 9** Unbiased two-electrode performance of TiO$_2$|CsPbBr$_3$|m-c|GS and TiO$_2$|CsPbBr$_3$|m-c|GS|WOC. **a** Chronoamperometric trace of TiO$_2$|CsPbBr$_3$|m-c|GS recorded in a two-electrode cell. **b** Chronoamperometric trace of TiO$_2$|CsPbBr$_3$|m-c|GS|WOC recorded in a two-electrode cell (0.1 M KNO$_3$ with pH adjusted to 2.5 with H$_2$SO$_4$, with no external bias voltage applied)

**Ir catalyst preparation**. The iridium precursor [Cp*Ir(pyalc)Cl] was synthesised according to previous literature reports[59]. Forty-eight milligrams of the precursor (0.1 mmol) was added to 20 ml of DI water in an open beaker and a clear orange solution was stirred vigorously for 5 min. Under constant stirring, 540 mg (2.5 mmol) of NaIO$_4$ were added to the beaker and the dark blue solution was allowed to stir overnight at room temperature. A piece of GS was floated on the unstirred, blue Ir-WOC solution overnight to bind the catalyst. It was then rinsed with DI water and left to dry at room temperature.

**Device characterisation**. Powder X-ray diffraction (XRD) patterns were collected using a Bruker Advance D8 X-ray diffractometer with a Cu Kα source. Measurements were taken from 2θ values in the range 5–80°.

Thin film UV-vis optical transmission and reflectance measurements were performed on a Perkin-Elmer Lambda 750S UV/Vis spectrometer, from 900 to 300 nm. The absorption coefficient was calculated as $\alpha = \ln\left(\frac{(1-R)^2}{T}\right)$.

The cross-sectional microscopy morphology was studied using a JEOL SEM6480LV scanning electron microscope (SEM) (20 kV acceleration voltage and a magnification of 5000) and a JEOL JSM-6301F Field emission scanning electron microscope (FESEM) for high resolution imaging.

Solar cell JV curves were measured using Keithley 2601A potentiostat, under simulated sunlight AM 1.5 G (100 mW cm$^{-2}$) with a solar simulator Class AAA with a HMI Lamp (200W/70V). A WPVS reference cell (Fraunhofer ISE) in accordance with international standard requirements of IEC 60904-2 was used to calibrate the light. The cell was held at 1.5 V under illumination for 5 s before scanning in reverse. The PV performance was not confirmed from independent certification laboratories. The voltage was swept from 1.5 to 0 V and back to 1.5 V at 100 mV s$^{-1}$.

The PEC performances of the devices were investigated in aqueous solutions 0.1 M KNO$_3$, with pH adjusted by using H$_2$SO$_4$ or KOH. When specified, a buffered solution of K-borate at pH 9 was used. The measurements were carried out in a three-electrode system with Pt as a counter electrode and Ag/AgCl as a reference electrode. The PEC electrolyte was exposed to air during measurement. The measured potentials versus Ag/AgCl were then converted to the RHE scale using the Nernst equation $E_{RHE} = E_{Ag/AgCl} + 0.059pH + 0.205$. The PEC performance, such as LSV and chronoamperometry, was measured with a Compactstat IVIUM potentiostat under simulated solar illumination (AM 1.5 G filtered 100 mW cm$^{-2}$) with 300 W Xe source (Lot-QD) from the back side. The intensity of the light at the working electrode position was measured by certified and calibrated SEL033/U photodetector (International Light Technologies). UV or IR light reaching the FTO-coated glass TEC7 is not further filtered but the latter glass may do some UV filtering (absorption spectra of FTO and FTO|TiO$_2$ are in Supplementary Fig. 28). LSV curves were measured at a scanning rate of 20 mV s$^{-1}$ and chronoamperometry experiments were conducted at 1.23 V$_{RHE}$.

IPCE measurements were performed from 300 to 900 nm with the same light source passing through a monochromator (MSH-300F LOT QuantumDesign) without the AM 1.5 G filter at an applied bias of 1.23 V$_{RHE}$. The intensity of the monochromatic light was measured by a SEL033/U photodetector (International Light Technologies).

OCP measurement were performed in a 3-electrode PEC cell. The voltage was measured versus the reference electrode Ag/AgCl when the current flowing through the system was set to be 0. The photovoltage was calculated by subtracting the OCP measured in the dark and under 1 sun illumination (AM 1.5 G filtered 100 mW cm$^{-2}$).

The O$_2$ evolution was probed using a compact fibre-optic oxygen metre (FireStingO$_2$) using a robust oxygen probe (XB7-546-206) in the three-electrode system previously purged with N$_2$ in a gas-tight quartz cell. TDIP temperature sensor was used to give automatic compensation for minor fluctuation in the PEC cell temperature. The faradaic efficiency was calculated by dividing the experimental O$_2$ produced and the theoretical O$_2$ calculated from

chronoamperometric traces, assuming that O$_2$ formation liberates four electrons. The amount of oxygen dissolved in the liquid was calculated according to Henry's law ($K_H = 769.23$ atm M$^{-1}$) and added to the amount detected in the headspace.

CV scans of GS and GS|WOC were performed using an AUTOLAB potentiostat in H$_2$SO$_4$ solution (pH 3.5) with 0.5 M KNO$_3$ at a scan rate of 20 mV s$^{-1}$. Ag/AgCl and Pt were used as reference and counter electrodes, respectively.

Ultimate stress and strain was measured using a 50 kN Instron Instrument and samples prepared according to the standard for tensile testing of polymer thin films. Sample was 75 mm long and ~6 mm wide along the working region of the tensile test sample. Stress calculated according to stress = force/area.

## Data availability

All relevant data are available from the authors. All data associated with the paper have been deposited online and can be freely and permanently accessed at https://doi.org/10.15125/BATH-00581[60].

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

## Acknowledgements

This work was supported by the EPSRC Centre for Doctoral Training in Sustainable Chemical Technologies (EP/L016354/1) at the University of Bath. I.P. thanks the European Union's Horizon 2020 research and innovation programme H2020-MSCA-CO-FUND-2014 (# 665992, MSCA FIRE: Fellows with Industrial Research Enhancement), U.H. thanks the Royal Society for a University research Fellowship (UF160458) and S.E. and S.K. thank the EPSRC for financial support (EP/R035407/1). J.B. and T.W. thank Self-Assembling Perovskite Absorbers Cells Engineered into Modules (SPACE-Modules) (EP/M015254/2) and the SPECIFIC Innovation and Knowledge Centre (EP/N02083/1). All authors thank the MAS staff at the University of Bath Mrs Ursula Potter, Dr Philip Fletcher and Ms Diana Lednitzky for experimental assistance in SEM acquisition and Dr Olivier Camus for Hg porosimetry. The authors thank Prof Steve Tennisson for valuable advice on carbon materials for this project.

## Author contributions

I.P., U.H., S.E. and P.J.C. conceived and planned the experiments. I.P. prepared the perovskite solar cells and photoanodes and conducted the experimental research. M.R. helped with IPCE. and oxygen-evolution measurements. S.K. performed the tensile strength measurements. E.V.S. and U.H. prepared the iridium WOC and developed the surface functionalisation. J.B. and T.M.W. provided advice on the mesoporous carbon material used in this project and its characterisation and contributed to the analysis and discussion of the results. IP wrote the manuscript with support of U.H., S.E. and P.J.C. who all contributed to analysing the data and discussing the results.

## Additional information

**Competing interests:** US Patent 9/790/605 by U.H. et al. contains intellectual property described in this article. The other authors declare no competing interest.

