## [Peer Review File · Nature Communications]

Reviewers' comments:

Reviewer #1 (Remarks to the Author):

The manuscript describes a facile approach to improve the stability of the halide perovskite based photoanodes for water splitting. When applied the graphite sheet on the top of electrode, by virtue of the high chemical inertness and the distinct hydrophobicity, the photoanodes shows an record stability, which is proved by the chronoamperometric tests and the complementary PV tests. Furthermore, through water oxidation catalyst (WOC) loading the onset potential can be cathodically shifted. The paper will be of interest to the wide readership of Nature Communications. However, there are some issues of concern that need to be addressed.

1. The cross-section image is of low quality. I cannot observe the TiO₂ and perovskite region clearly. Please retake it and indicate the interface. In addition, the cross-sectional SEM of the TiO₂ | CsPbBr₃ | m-c | GS is missing. The authors should provide the SEM result of the electrode to illustrate the structure.

2. Formation of the WOC and functionalisation of the GS surface can enhance the activity. However, conclusive evidence for the presence of the WOC should be given. XPS or Elemental distribution of Ir may be a better way to prove this. Furthermore, the PH of solution in the figure 6 and figure 2 is different. In figure 7, the relationship between the PH of the solution and onset potential and OCP has been pointed out. The Ir-WOC cannot always lead to an improvement, which is ascribed to high activity of the Ir-WOC at low pH and the deactivation of the Ir-WOC at high pH. As a result, the author may put emphasis on the functionalisation of the GS surface instead of the further improvement.

3. Despite that graphite is good conductor, the thickness of the protective should not be so large, which marks the light-induced charges hard to transport to the semiconductor/liquid interface. The author should provide the thickness dependence for the PEC/PV performance as well as the stability. Will increasing the thickness enhance its stability?

4. Why there is a drop in the IPCE curve around 450 nm?

5. The author should explain the reason of increasing photocurrent during the measurement. Will it be the influence of temperature after long-term exposure or the soaking of electrolyte during the measurement?

6. Why the faradic efficiency for the first 30 min is so poor?

7. The photocurrent is positive even at potential of -0.5 V vs. RHE according to Fig. 6. How can the electrode oxide water even at this potential?

8. Now, the world-record efficiency performance is 23.7%.

Reviewer #2 (Remarks to the Author):

This paper reports on photoelectrochemical water splitting based on CsPbBr₃ photoanode protected by a mesoporous carbon and graphite sheet materials with surface-attached Ir-based catalyst for oxygen evolution.

Authors have conducted comprehensive fabrication and characterization of the perovskite device. Novelty of the paper includes use of the high-bandgap CsPbBr₃ material, use of mesoporous carbon as protection layer, and somewhat longer stability compared to previous reports.

In spite of all the merits, as authors appropriately summarized in Introduction, there have already been quite a few reports on 'direct' photoelectrochemical water splitting using perovskites. Different labs have used different materials to protect the moisture-sensitive perovskite including a thin Ni layer, a carbon nanotube/polymer composite, and Field's metal. Even though there are some improvements in this work, this reviewer finds that this paper does not demonstrate novelty and breakthrough that are required to be publishable in a high-impact journal like Nature Communications.

In addition, this reviewer would like to add a few comments:

- As it seems that authors used the high-bandgap CsPbBr₃ to achieve an unbiased solar water splitting, it is recommended that authors perform a two-electrode measurement and calculate the STH (solar-to-hydrogen) efficiency.
- In Fig. 4, the long-term stability of the device was measured in a benign acidity (pH 6.8) However, practical solar water splitting should operate either in strong acid or basic condition. Authors should at least compare device stability in different pH's.

Reviewer #3 (Remarks to the Author):

This work is a thorough study on the performance of CsPbBr₃ photoanodes for water oxidation. The authors propose a very simple and versatile device architecture, which results in an improved stability under use. Yet, there are a few issues which need to be addressed before acceptance.

1. The introduction lacks many examples of perovskites used for water splitting. The authors claim that only one example of a perovskite solar cell – photoanode tandem exists, whereas at least a dozen examples are known by now. Moreover, the authors claim that only the CH₃NH₃PbI₃ perovskite has been investigated so far for water splitting, without taking the works of Grätzel (Adv Energy Mater 2016), Yang (J Mater Chem A 2017) and Reisner (Adv Energy Mater 2018) on mixed-cation mixed-halide perovskites into account. Same can be said about the stability, which is comparable to other current state-of-the-art PEC tandem systems. I recommend more caution in respect to those claims and the authors referring to these papers for a more comprehensive overview of existing literature.

2. The authors explain the large negative current observed at cathodic potentials in Fig. S5c as caused by oxygen reduction. Were the LSVs recorded in a closed, or in an open container? If the curves were recorded in a closed PEC cell, why did the authors not purge it with an inert gas beforehand. Gas analysis of the resulting products and faradaic yield can help differentiate between oxygen reduction, device degradation, or an eventual hydrogen evolution. Is the LSV scan recorded in forward or backward direction? Why is there no similar reductive curve in Fig. S8?

3. The difference between the IPCE of the devices (see Fig. S10) with or without the graphite sheet is much higher than the difference between photocurrents observed from the LSVs (Fig. S7c and 3c). Can the authors integrate the IPCE data to obtain the theoretical photocurrents, and compare those to the recorded ones? At which potential is the IPCE data recorded, 1.23V vs RHE?

4. The authors state that "the TiO₂|CsPbBr₃|m-c|GS|WOC cell achieved a photocurrent which was about five times higher than that of the unfunctionalised TiO₂|CsPbBr₃|m-c|GS cell at 0V vs RHE (Fig.

6c)". If one looks purely at those photocurrents, then the onset potential of the water oxidation for the device with the WOC should be at roughly -0.8V vs RHE, which is unrealistic even with the wide band-gap CsPbBr₃ perovskite, and is inconsistent with the onset potentials calculated in Fig 7b and S17. By looking at the large hysteresis of the CV scans in Fig. S14, one might suspect that the observed photocurrent originates from capacitive charging of the carbon encapsulant, and by oxidising another redox species of the catalyst, respectively. Intrinsic perovskite device hysteresis also adds on top of that (Fig. 6b).

Can the authors assign the other waves below 0.8V vs Ag/AgCl in Fig. S14 to any Ir species? The authors should present cyclic voltammetry scans of the photoanodes, not just LSVs, to show whether hysteresis/capacitance/catalyst redox chemistry plays a role in their observed photocurrents. I especially suggest running chronoamperometry at 0V vs RHE for both electrodes, to see whether this photocurrent is sustained over time, and if any O₂ gas is produced. If the photoanode with a WOC can indeed sustain O₂ evolution at 0V vs RHE, then this work has the potential to become a milestone for the water oxidation community; otherwise, Fig. 6c and its description are just misleading. This point needs to be properly addressed in order to grant publication.

5. Can the authors split the box plots in Fig S16a in photovoltaic devices used for the photoanodes with and without the WOC? Even if the perovskite batches may have been different (which is not clear at the moment), the average water oxidation photocurrents are equal. Does that point towards kinetic water oxidation limitations?

Minor issues:

6. Could the authors add some arrows on Fig. S6b to indicate the three electrodes? The setup is not so clear from the current angle.

7. Fig. S7: It is confusing that the authors zoom-in on the chopped photocurrent in frame c, but they average the light points in frame b. Maybe just show the raw data in frame b as well.

8. Why do the authors not show the entire chronoamperometric trace from Fig. S12a in Fig. 4a, instead of only arbitrarily showing a part of it?

9. Many experiments are run at different pH values (3.5, 6.8, 9, 12.5, then 3.1, 4.3, 7, 10.6 and 12.9 in Fig. S17), with different electrolyte salts (K borate, KNO₃, KOH, some unnamed). While this indicates work conducted by several collaborators independently from each other, could the authors try to give a reason for not using a fixed buffer / buffer range? Can the authors accurately describe all buffers used for the respective experiments?

10. How is the open circuit potential (OCP) defined? Is it the potential observed vs Ag/AgCl?

11. Conclusions: I think the word "oxidation" is missing from "above 70% for direct light-driven water, showing".

Response to Reviewers' comments

Reviewer #1:

The manuscript describes a facile approach to improve the stability of the halide perovskite based photoanodes for water splitting. When applied the graphite sheet on the top of electrode, by virtue of the high chemical inertness and the distinct hydrophobicity, the photoanodes shows a record stability, which is proved by the chronoamperometric tests and the complementary PV tests. Furthermore, through water oxidation catalyst (WOC) loading the onset potential can be cathodically shifted. The paper will be of interest to the wide readership of Nature Communications. However, there are some issues of concern that need to be addressed.

We thank the reviewer for the positive evaluation of our manuscript, acknowledging the originality and significance of our results. In the following, we have tried to address all issues raised.

1. The cross-section image is of low quality. I cannot observe the TiO₂ and perovskite region clearly. Please retake it and indicate the interface. In addition, the cross-sectional SEM of the TiO₂|CsPbBr₃|m-c|GS is missing. The authors should provide the SEM result of the electrode to illustrate the structure.

As requested, we have now measured the cross-sectional SEM image of TiO₂|CsPbBr₃|m-c sample with a field emission SEM to obtain higher resolution images. The image now clearly shows the TiO₂|CsPbBr₃ interface (Figure R1). The compact TiO₂ measured about 50 nm and the CsPbBr₃ layer about 350 nm, with large grains growing vertically throughout. **The new SEM image has been added to the revised manuscript (Figure 1).**

Figure R1. Cross sectional SEM image of TiO₂|CsPbBr₃|m-c architecture, including magnification of the TiO₂|CsPbBr₃|m-c interface.

We also tried to take a cross-sectional SEM image of TiO₂|CsPbBr₃|m-c|GS but found it very difficult to cleanly cut the sample (by different techniques) once the GS had been applied due to its stiffness and the underlying adhesive. Thus, no meaningful images of TiO₂|CsPbBr₃|m-c|GS could be

obtained; but the structure of the cell underneath the graphite sheet should be very similar to that shown for identically prepared cells without graphite sheet.

2. Formation of the WOC and functionalisation of the GS surface can enhance the activity. However, conclusive evidence for the presence of the WOC should be given. XPS or Elemental distribution of Ir may be a better way to prove this. Furthermore, the PH of solution in the figure 6 and figure 2 is different. In figure 7, the relationship between the PH of the solution and onset potential and OCP has been pointed out. The Ir-WOC cannot always lead to an improvement, which is ascribed to high activity of the Ir-WOC at low pH and the deactivation of the Ir-WOC at high pH. As a result, the author may put emphasis on the functionalisation of the GS surface instead of the further improvement.

We have now used EDX on GS|WOC samples to determine the elemental composition of the functionalised GS surface. Figure R2 shown below is the top-view SEM image of the functionalised surface and its corresponding EDX analysis.

Figure R2. Top-view SEM image and EDX chemical composition analysis of the surface of GS functionalised with the Ir-based WOC

The main elements found on the surface are carbon, oxygen and iridium, providing direct evidence for the presence of the catalyst in addition to our electrochemical and photoelectrochemical data. Due to the nature of the GS synthesised from oriented graphite flakes, its surface is smooth and compact with few defects and oxidic edges the catalyst can bind to (as shown in ref. [1]). Thus, the distribution of the Ir-WOC is not as uniform and the loadings are lower than reported previously for porous metal oxide architectures (as used in ref. [1]). SEM images and EDX analysis of different areas of the surface can be found in the new version of the SI (Figure S19).

Figure R3 shows additional CV scans of GS|WOC measured in aqueous solutions at different pH. As previously reported for the Ir-WOC in solution [2] and on ITO surfaces [1], there is quasi-reversible voltammogram (close to 1 V vs. RHE) which is visible in acidic media. The response is due to the oxidation and re-reduction of the Ir^{III}/Ir^{IV} catalyst. The onset of the water oxidation starting at about

1.4 V_{RHE} at pH 2.6 shifts to higher potential with increasing pH. CV scans of GS|WOC electrodes are now shown in the main manuscript (Figure 5b) and in the SI (Figure S20).

Figure R3. (top) CV scan of GS and GS functionalised with the Ir-WOC measured in 0.1 M KNO_3 at pH 2.6, with scan rate of 50 mVs^{-1} . The inset shows a close-up of the potential axis, where the $\text{Ir}^{\text{III}}/\text{Ir}^{\text{IV}}$ couple can be seen. (bottom) CV scans of GS|WOC measured in 0.1 M KNO_3 solution at different pH (adjusted with H_2SO_4 and KOH), scan rate 50 mVs^{-1}

In order to assess the kinetic acceleration that the catalyst brings to the overall device at different pH more clearly, overpotentials for oxygen evolution on bare GS and Ir-WOC functionalised GS were measured and compared. Figure R4 clearly shows the lower kinetic overpotential that the addition of the Ir-WOC affords.

Figure R4. GS|WOC and GS overpotential as a function of the pH of solution

Figure R4 and relative description have been added to the revised manuscript (Figure 5c).

3. Despite that graphite is good conductor, the thickness of the protective should not be so large, which marks the light-induced charges hard to transport to the semiconductor/liquid interface. The author should provide the thickness dependence for the PEC/PV performance as well as the stability. Will increasing the thickness enhance its stability?

Initially, we used a 25 μm thick GS because it was the thinnest available on the market. Figure R5 below shows a comparison of JV curves of $\text{TiO}_2|\text{CsPbBr}_3|\text{m-c}$ with (black) and without (red) GS applied on the surface. Even though the protective GS was much thicker than the absorber layer, the photovoltaic properties of the complete solar cells were only slightly affected, showing that light induced charges must be effectively transported across the entire device. To clarify this point we added this figure to the SI (figure S12).

Figure R5. JV curve of TiO₂/CsPbBr₃/m-c before (black) and after (red) applying a 25 μm GS on top of the m-c surface

As suggested by the reviewer, we have now also tested a 70 μm thick GS (GS70). As it can be seen from Figure R6, charges are still extracted even with an almost three times thicker protection layer.

The same figure has been also added to the SI (Figure S17).

Figure R6. JV curve of TiO₂/CsPbBr₃/m-c before (black) and after (green) applying a 70 μm GS on top of the m-c surface

In order to assess whether a thicker GS may bring about enhanced stability in light-driven water splitting, we measured the performance of TiO₂/CsPbBr₃/m-c/GS70 in 0.1 M KNO₃ pH 7 under continuous illumination and 1.23 V_{RHE}. We were pleased to observe a substantial increase in lifetime for the photoanode (Figure R7). The generated photocurrent increased by over 30% of its initial

value during the first 3 hours and then stabilised to a photocurrent of 2.4 mAcm^{-2} after 5 hours. In total, over 2 mAcm^{-2} were maintained for about 30 hours before the device failed due to the epoxy and resin sealing materials disintegrating (leading to water ingress that degraded the CsPbBr_3). Unlike the results with GS25, the GS70 was still intact at the end of the experiment and had not reached its fracture point yet. This clearly indicates that thicker GS protection layers, along with better sealing materials, are able to extend the lifetime of these composite photoanodes even further.

These new results have been added to the manuscript (Figure 4 and Figure S18) by including the stability curve and by discussing the improvements in the lifetime observed when thicker GS protection layers were used.

Figure R7. PEC stability of $\text{TiO}_2/\text{CsPbBr}_3/m\text{-c}/\text{GS70}$ measured in 0.1 M KNO_3 ($\text{pH } 7$) at $1.23 \text{ V}_{\text{RHE}}$ and photograph of the electrodes in the PEC cell after the 35 hours measurement

4. Why there is a drop in the IPCE curve around 450 nm?

We have now carefully repeated the experiment with better light intensity calibration, and indeed obtained a more consistent IPCE curve, the new curve is reproduced below (Figure R8). The new IPCE curve is now shown in Figure 3a of the revised manuscript.

Figure R8. IPCE of $\text{TiO}_2/\text{CsPbBr}_3/\text{m-c}/\text{GS}$ measured in aqueous solution (pH 7) at $1.23 V_{\text{RHE}}$ under monochromatic light irradiation

5. The author should explain the reason of increasing photocurrent during the measurement. Will it be the influence of temperature after long-term exposure or the soaking of electrolyte during the measurement?

We have investigated both possibilities and found the increase in temperature during the experiment to be negligible ($<1^\circ\text{C}$). On the other hand, we observed that the GS material used to encapsulate the moisture-sensitive perovskite layer did experience changes in its mechanical properties when exposed to aqueous electrolyte (as previously shown in Figure S13 in the original SI). We believe that the decrease in stiffness observed may be a result of swelling during soaking that can expose a higher surface area to the electrolyte, thereby increasing water oxidation performance. To further verify the effect of soaking, we tested the performance of a $\text{TiO}_2/\text{CsPbBr}_3/\text{m-c}/\text{GS}$ photoanode which was subjected to intermittent electrolyte soaking in the dark. The results in Figure R9 show that the photocurrent density increased every time the photoanode was illuminated again, suggesting that electrolyte soaking is chiefly responsible for causing the initial increase in photocurrents. These results have been added to the revised version of the SI (Figure S16) and discussed in the revised manuscript.

Figure R9. Chronoamperometric response of $\text{TiO}_2/\text{CsPbBr}_3/m\text{-c}/\text{GS}$ recorded in 0.1 M KNO_3 solution ($\text{pH } 7$) with an applied potential of $1.23 \text{ V}_{\text{RHE}}$. Between each measurement the device was kept in solution for 1 hour in the dark at open circuit

6. Why the faradaic efficiency for the first 30 min is so poor?

The Faradaic efficiency was calculated by dividing the actual concentration of O_2 detected in the headspace of a sealed photoelectrochemical cell by the theoretical concentration of O_2 produced based on the photocurrents measured. Thus, there is a delay before the sensor in the headspace measures the oxygen concentration. While current flow occurs almost immediately upon illumination, in order to reach the sensor any O_2 produced by the photoanode in solution has to detach from the GS surface, saturate the electrolyte solution, and then diffuse out into the headspace and through the sensor membrane. This causes a delay of several minutes to hours (depending on volume ratios and amount of O_2 produced) between the generation of O_2 and its actual detection, an effect that is known in the field and has been reported in prior literature (see for instance ref. [3]). Once the system has reached steady state, the true Faradaic efficiency of the device can be assessed (82.3 % in our case).

To clarify the faradaic efficiency results, text that explains the delay between the generation and the detection of O_2 has been added in the SI in section S3.1.

7. The photocurrent is positive even at potential of -0.5 V vs. RHE according to Fig. 6. How can the electrode oxide water even at this potential?

We thank the reviewer for pointing out this important detail. To understand which component of our composite photoanode was producing photocurrents at these low potentials, we prepared and measured TiO_2 |m-c|GS and CsPbBr_3 |m-c|GS as control devices.

Figure R10. LSV under chopped solar light illumination at different applied potentials of TiO_2 |m-c|GS and CsPbBr_3 |m-c|GS

This comparison in Figure R10 clearly showed the photocurrents seen at negative potentials appear to originate from the CsPbBr_3 absorber material. We agree with the reviewer that it is unlikely to be the result of photocatalytic oxygen evolution, and propose it is a capacitive charging effect caused by ion movement within the perovskite layer as the light is switched on and off – such responses are well known for lead halide perovskite cells which show substantial JV curve hysteresis due to ion movement. To investigate further, the photocurrent for a complete TiO_2 | CsPbBr_3 |m-c|GS device was measured under both continuous and chopped illumination. Under chopped illumination a charging current caused by ion reorganisation should flow each time the light is switched on or off. Under continuous illumination the light induced charging current should decay in the first few seconds of the measurement. Figure R11 shows the photocurrent measured for the same device under continuous and chopped illumination. Consistent with this hypothesis, we observed very low currents at negative potentials under continuous illumination (Figure R11 right) that started to rise steeply above $0.5 V_{\text{RHE}}$ when true oxygen evolution set in. In contrast, under chopped illumination a larger positive photocurrent is seen at negative potentials. Importantly, the interpretation of photocurrents below $0.5 V_{\text{RHE}}$ as capacitive charging due to ion movement, and above $0.5 V_{\text{RHE}}$ as true

oxygen evolution is consistent with our original onset potential analysis of the photoanodes (Figure 7b and S17 of original manuscript and S1). As the results showed LSV curves measured under continuous illumination to be more representative of the real currents that can be achieved, we have now replaced all LSV curves measured under chopped illumination with curves measured under continuous illumination in the revised document. LSV curves of both $\text{TiO}_2/\text{CsPbBr}_3/\text{m-c}/\text{GS}$ and $\text{TiO}_2/\text{CsPbBr}_3/\text{m-c}/\text{GS}/\text{WOC}$ measured under continuous illumination have been added to the revised manuscript (Figure 2c and Figure 6b).

Figure R11: LSV of $\text{TiO}_2/\text{CsPbBr}_3/\text{m-c}/\text{GS}$ under chopped and continuous illumination at different applied potentials (0.1 M KNO_3 , pH 4.3).

8. Now, the word- record efficiency performance is 23.7%.

We thank the reviewer for pointing out the latest development, which we have now included in the revised introduction.

Reviewer #2:

This paper reports on photoelectrochemical water splitting based on CsPbBr₃ photoanode protected by a mesoporous carbon and graphite sheet materials with surface-attached Ir-based catalyst for oxygen evolution. Authors have conducted comprehensive fabrication and characterization of the perovskite device. Novelty of the paper includes use of the high-bandgap CsPbBr₃ material, use of mesoporous carbon as protection layer, and somewhat longer stability compared to previous reports. In spite of all the merits, as authors appropriately summarized in Introduction, there have already been quite a few reports on 'direct' photoelectrochemical water splitting using perovskites. Different labs have used different materials to protect the moisture-sensitive perovskite including a thin Ni layer, a carbon nanotube/polymer composite, and Field's metal. Even though there are some improvements in this work, this reviewer finds that this paper does not demonstrate novelty and breakthrough that are required to be publishable in a high-impact journal like Nature Communications.

We thank the reviewer for their thorough and positive assessment of our work. While it is true that different labs have reported different strategies towards protecting the moisture-sensitive perovskite for applications in direct water splitting, none of the approaches have proven very successful so far. As laid out in our introduction, devices protected by Ni layers achieved a continuous photocurrent for only ten minutes in aqueous solution. Carbon nanotube/polymer composites have been used as hole transporting layer in solar cells, which showed unchanged photovoltaic properties after the cell was placed under running tap water for 60 seconds [4]. When the same composite layer was used as protecting layer and integrated with the Ni encapsulation technique, the operational stability was extended to up to tens of minutes [5]. Other reports use Field's metal applied on perovskite-based photocathodes for water reduction – these metal coated electrodes showed comparable stabilities to the ones we reported in the original manuscript. However, the 2 electron proton reduction is both kinetically and thermodynamically much more facile than 4 electron water oxidation, and Field's metal is very expensive due to its 50 %wt indium content.

Thus, none of the literature reports we are aware of come close to the performance we were able to achieve with a cheap and readily fabricated architecture such as mesoporous carbon and graphite sheets (we now have reached a record lifetime of 34 hours under continuous illumination in aqueous electrolyte). In addition, we demonstrate a facile and efficient way of functionalising the electrolyte-facing photoanode surface with a water-oxidation catalyst that improves the

performance of the device further. This important step has not been demonstrated for any perovskite-based photoelectrodes so far. We therefore believe our work to constitute a significant advancement suitable for publication in a high-impact journal such as *Nature Communications*.

1. As it seems that authors used the high-bandgap CsPbBr₃ to achieve an unbiased solar water splitting, it is recommended that authors perform a two-electrode measurement and calculate the STH (solar-to-hydrogen) efficiency.

In the original manuscript we wished to state that it *might* be possible to do unbiased water splitting with CsPbBr₃ devices, we did not mean to suggest that we were carrying out unbiased water splitting in these measurements. **In order to avoid confusion about the possibility of performing unbiased water splitting with CsPbBr₃ we have now amended these sections (highlighted text at page 15 of the revised manuscript).**

CsPbBr₃ has a bandgap of 2.3 eV in the visible range, resulting in photovoltages of about 1.3 V (as measured and shown in in Fig 7c of the original manuscript). While this value is above thermodynamic for the electrolysis of water (1.23 V), it leaves little overpotential for driving the kinetics. Typically another 0.3 V (or more, depending on the materials used) are required to produce measurable quantities of H₂ and O₂ from H₂O. Thus, while in principle it is thermodynamically possible to perform unassisted water splitting using CsPbBr₃ based photoanodes, in the original manuscript we applied a bias potential > 0.5 V_{RHE} to speed up charge separation and generate significant photocurrents and measurable O₂ evolution.

Nevertheless, following the requests from reviewers 2 and 3 (see below), we have now performed a two-electrode measurement without applying any external bias between the working and counter electrode. Figure R12 shows the responses obtained from TiO₂|CsPbBr₃|m-c|GS (left) and TiO₂|CsPbBr₃|m-c|GS|WOC (right) photoanodes in water, respectively. The unbiased photocurrent density generated by TiO₂|CsPbBr₃|m-c|GS was barely measurable due to the sluggish kinetics of oxygen evolution at the ~1.3 V provided by the illuminated perovskite. However, with the addition of the water-oxidation catalyst more meaningful photocurrents, up to three times higher, could be detected.

Figure R12. Chronoamperometric trace recorded in a two-electrode cell in 0.1 M KNO_3 (pH adjusted to 2.5 with H_2SO_4) with no external bias voltage applied.

These measurements support our statement that unassisted water splitting is possible with CsPbBr_3 based photoanodes. However, when we tried to measure the amount of oxygen produced under the conditions shown in Figure R12 using an oxygen probe in de-aerated solution, the amount of O_2 produced was below the detection limit of our probe. Figure R13 shows the photocurrent generated under these unbiased conditions together with the predicted amount of O_2 production. The amount of oxygen generated after 2 hours is 28 nmol; which as stated above is below the detection limit of our most sensitive probe. These results have been added to the revised SI (Figure S27) and to the revised manuscript (Figure 9).

Figure R13. Chronoamperometric trace recorded in a two electrode cell in 0.1 M KNO_3 (pH adjusted to 2.5 with H_2SO_4) with 0 V applied for 2 hours and predicted O_2 production (assuming that O_2 formation liberates 4 electrons.)

2. In Fig. 4, the long-term stability of the device was measured in a benign acidity (pH 6.8) However, practical solar water splitting should operate either in strong acid or basic condition. Authors should at least compare device stability in different pH's.

We would like to point out that traditional electrolyzers have indeed been developed to work best with either acidic or alkaline solutions. However, this is dictated by the proton exchange membranes separating the cathodic H_2 evolution and the anodic O_2 evolution half-cells, which require either an excess of H^+ or OH^- to work efficiently under high-load conditions. In solar water splitting current densities and gas evolution rates are much lower, so there is less of a need to operate under extreme pH conditions. In fact, neutral (sea or ground) water would arguably be the best media for distributed solar energy conversion based on photoelectrochemical water splitting [6],[7]. This is also an advantage from a device perspective, as many of the components used in photoelectrodes (semiconductors, electrocatalysts, sealants, etc.) are unstable under harsh pH conditions. Therefore, it is important to test solar water-splitting devices at near-neutral pH and the fact that our composite photoanode can operate at near-neutral pHs is actually advantageous.

Nevertheless, following the reviewer's request, we investigated our device stability in acidic and alkaline conditions as well (Figure R14). While acidic conditions led to shorter lifetime of about 8 hours, we achieved a new record 34 hour stability at pH 12.5. In all cases the end of life of the devices was caused by delamination and fracture of the 25 μm GS. Figure R14 has been added to the revised SI (Figure S13) and we expanded the discussion in the text of the revised manuscript.

Figure R14. Chronoamperometric trace of $\text{TiO}_2/\text{CsPbBr}_3/m\text{-c}/\text{GS}$ recorded at an applied potential of $1.23 V_{\text{RHE}}$. 0.1 M KNO_3 electrolyte solution with pH adjusted with H_2SO_4 and KOH , under continuous solar light irradiation ($\text{AM } 1.5 \text{ G}, 100\text{mWcm}^{-2}$).

Reviewer #3:

This work is a thorough study on the performance of CsPbBr₃ photoanodes for water oxidation. The authors propose a very simple and versatile device architecture, which results in an improved stability under use. Yet, there are a few issues which need to be addressed before acceptance.

We thank the reviewer for acknowledging the novelty and impact of our work, and hope that all issues standing in the way of acceptance have now been fully addressed.

1. The introduction lacks many examples of perovskites used for water splitting. The authors claim that only one example of a perovskite solar cell – photoanode tandem exists, whereas at least a dozen examples are known by now. Moreover, the authors claim that only the CH₃NH₃PbI₃ perovskite has been investigated so far for water splitting, without taking the works of Grätzel (Adv Energy Mater 2016), Yang (J Mater Chem A 2017) and Reisner (Adv Energy Mater 2018) on mixed-cation mixed-halide perovskites into account. Same can be said about the stability, which is comparable to other current state-of-the-art PEC tandem systems. I recommend more caution in respect to those claims and the authors referring to these papers for a more comprehensive overview of existing literature.

We would like to clarify that different configurations for solar water splitting using perovskite materials have been explored. For the purpose of direct photoelectrochemical water splitting, the cell typically includes either a single photoelectrode connected to a metal counter-electrode (half-cell), or two coupled photoelectrodes that work as photoanode and photocathode (tandem cell). In this work, we used the first type of configuration, where a CsPbBr₃-based device was used as a photoanode in solution coupled with a platinum counter electrode. Therefore, we decided to focus our introduction on the available examples that used such “half-cell” configuration with halide perovskite based materials. The Yang paper mentioned (J Mater Chem A 2017) reported a study of a CH₃NH₃PbI₃-based photoanode, which we cited in our original manuscript (as reference 25). Reisner’s recent report (Adv Energy Mater 2018) focused on a tandem cell, using a perovskite based photocathode following the same encapsulation method described previously by the same group, which we cited in our original manuscript as reference 26. **For completeness sake, we have now added this new reference to the revised version of the manuscript (reference 30).**

A third possible configuration is when water splitting is driven by an external photovoltaic solar cell (the so-called PV-electrolyser). Although we did mention the first example of such a setup using a

perovskite solar cell published by Grätzel in 2014 (original reference 21), we initially decided not to expand on these types of devices because as the perovskite material is not in direct contact with the aqueous electrolyte, such systems do not face the same challenges approaches which use submerged photoelectrodes (as we do here). However, to give a more comprehensive overview, we have now added these other examples that employ the PV-electrolyser configuration to our introduction (references 22-24 and).

2. The authors explain the large negative current observed at cathodic potentials in Fig. S5c as caused by oxygen reduction. Were the LSVs recorded in a closed, or in an open container? If the curves were recorded in a closed PEC cell, why did the authors not purge it with an inert gas beforehand. Gas analysis of the resulting products and faradaic yield can help differentiate between oxygen reduction, device degradation, or an eventual hydrogen evolution. Is the LSV scan recorded in forward or backward direction? Why is there no similar reductive curve in Fig. S8?

All LSV curves shown were recorded in forward scan direction in a container opened to air, thus they show reduction currents that flow due to oxygen reduction at cathodic potentials. Only the determination of the Faradaic Efficiency was performed in a closed cell purged with N₂ in order to accurately quantify any oxygen produced. Since photoanodes produce oxygen, we think it is more relevant to show LSV curves from unpurged systems (as it is indeed common in the literature). Our original Figure S8, showing the LSV of TiO₂|PbBr₂|m-c under chopped illumination, had the dark current subtracted from the photocurrent density data. Figure R15 below shows the original data including large negative current at cathodic potentials indicative of O₂ reduction as it is now reported in the revised SI (Figure S8).

Figure R15. LSV of TiO₂|PbBr₂|m-c measured under simulated solar light in a buffer solution with pH=9.

The O_2 reduction features seen with our composite photoanodes are ascribed to originate mainly from any uncovered surface areas of the underlying FTO glass, which is well known to display such reactivity [8][9][10]. All new samples (which are shown in Figure 2c and 6b of the revised manuscript) have now been prepared by carefully coating all exposed FTO areas with epoxy resin to avoid oxygen reduction taking place at the FTO electrode. Indeed, the dark currents at reducing potentials were considerably reduced (Figure R16; anodic currents at higher potentials were not affected).

Figure R16. LSV of $TiO_2/CsPbBr_3/m-c|GS$ and $TiO_2/m-c|GS$ in 0.1 M KNO_3 under chopped solar light irradiation with and without FTO exposed to the aerated aqueous electrolyte.

3. The difference between the IPCE of the devices (see Fig. S10) with or without the graphite sheet is much higher than the difference between photocurrents observed from the LSVs (Fig. S7c and 3c). Can the authors integrate the IPCE data to obtain the theoretical photocurrents, and compare those to the recorded ones? At which potential is the IPCE data recorded, 1.23V vs RHE?

IPCE data were indeed recorded at 1.23 V_{RHE} and we have now added this information to the experimental section and in the relative figure captions. The photocurrent density measured at 1.23 V_{RHE} for $TiO_2|CsPbBr_3|m-c$ photoanodes was about 0.4 $mAcm^{-2}$ (Figure S7), and the IPCE was about 10% (Figure S10). In contrast, the photocurrent density measured at 1.23 V_{RHE} for the $TiO_2|CsPbBr_3|m-c|GS$ photoanode was about 2.25 $mAcm^{-2}$ (Figure 2c of the original manuscript), and the IPCE 70% (Figure S10). Thus, the photocurrent density without the GS was 17% of that with the GS, and the IPCE without the GS was 14% of that with the GS. Therefore, there is no large difference between the IPCE values and the photocurrent values. As suggested we integrated the IPCE data to give the theoretical photocurrents (Figure R17), we find values very similar to the actual measurements. We thank the reviewer for suggesting this useful analysis, and have now added this data to the SI (Figure S11).

Figure R17. Integrated photocurrent densities of $TiO_2|CsPbBr_3|m-c|GS$ and $TiO_2|CsPbBr_3|m-c$ calculated by IPCE data measured at 1.23 V_{RHE} in aqueous solution at pH 9.

4. The authors state that “the $TiO_2|CsPbBr_3|m-c|GS|WOC$ cell achieved a photocurrent which was about five times higher than that of the unfunctionalised $TiO_2|CsPbBr_3|m-c|GS$ cell at 0V vs RHE (Fig. 6c)”. If one looks purely at those photocurrents, then the onset potential of the water oxidation for the device with the WOC should be at roughly -0.8V vs RHE, which is unrealistic even with the wide band-gap $CsPbBr_3$ perovskite, and is inconsistent with the onset potentials calculated in Fig 7b and S17. By looking at the large hysteresis of the CV scans in Fig. S14, one might suspect that the

observed photocurrent originates from capacitive charging of the carbon encapsulant, and by oxidising another redox species of the catalyst, respectively. Intrinsic perovskite device hysteresis also adds on top of that (Fig. 6b).

We thank the reviewer for pointing this out. As explained in our response to Q7 of reviewer 1 above, we have now carefully dissected the two different regimes of photocurrents observed as capacitive charging and ion movement within the composite device below $0.5 V_{\text{RHE}}$ (low photocurrents that oscillate under chopped illumination), and true water splitting above $0.5 V_{\text{RHE}}$ (higher photocurrents that don't vary with illumination mode but respond to pH changes and in the electrolytes as expected). Thus, our original method of using the tangent to the photocurrent curve at $> 0.5 V_{\text{RHE}}$ (where the photocurrent increases strongly) to calculate onset potentials and our comparison of relative photocurrent differences $> 0.5 V_{\text{RHE}}$ are still valid.

We have now repeated the measurements with freshly made devices and continuous illumination (see discussion in response to reviewer 1 above), and obtained clearer LSV curves of $\text{TiO}_2|\text{CsPbBr}_3|\text{m-c}|GS$ and $\text{TiO}_2|\text{CsPbBr}_3|\text{GS}|WOC$, indicating onset potentials of 0.65 and $0.55 V_{\text{RHE}}$, respectively (Figure R18). The manuscript has been updated to show these results in Figure 6b.

Figure R18. LSV of a $\text{TiO}_2|\text{CsPbBr}_3|\text{m-c}|GS$ and $\text{TiO}_2|\text{CsPbBr}_3|\text{m-c}|GS|WOC$ indicating the onset potential under illumination measured in aqueous solutions 0.1 M KNO_3 with pH adjusted to 2.5 with H_2SO_4 .

5. Can the authors assign the other waves below 0.8V vs Ag/AgCl in Fig. S14 to any Ir species? The authors should present cyclic voltammetry scans of the photoanodes, not just LSVs, to show whether hysteresis/capacitance/catalyst redox chemistry plays a role in their observed photocurrents.

We have now collected cyclic voltammetry data of the Ir-WOC on GS (see Figure R3), which do not show extraordinary GS charging currents, we do clearly see a redox wave for Ir-WOC around 1 V vs RHE under acidic conditions. As explained in our response to Q2 of reviewer 1, these redox events are known to originate from pre-catalytic Ir^{III}/Ir^{IV} redox features (see also [1]). We have explained this more clearly in the revised version of the manuscript (Figure 5b and highlighted text at page 12).

We have now also performed cyclic voltammetry scans of TiO₂|CsPbBr₃|m-c|GS|WOC devices in 0.1 M KNO₃ solution at different pH under 1 sun illumination. Under these conditions, the reduction wave for the Ir^{V/IV} reduction could be observed on the reverse scan (the Ir^{V/IV} oxidation is hidden within the catalytic wave in the forward scan), as previously reported for hematite photoanodes functionalised with the same Ir-WOC [11]. Just like the Ir-WOC on GS, the TiO₂|CsPbBr₃|m-c|GS|WOC devices also responded to changes in the solution pH as expected from the Pourbaix diagram of H₂O. The CV scans have been added to the revised SI (Figure S23).

Figure R19. CV scans of TiO₂|CsPbBr₃|m-c|GS|WOC devices measured in solution 0.1 M KNO₃ with pH adjusted to 2.5, 4 and 7 with H₂SO₄

6. I especially suggest running chronoamperometry at 0V vs RHE for both electrodes, to see whether this photocurrent is sustained over time, and if any O₂ gas is produced. If the photoanode with a WOC can indeed sustain O₂ evolution at 0V vs RHE, then this work has the potential to become a milestone for the water oxidation community; otherwise, Fig. 6c and its description are just misleading. This point needs to be properly addressed in order to grant publication.

We apologise as we did not mean to be misleading and in the original manuscript we simply wished to suggest that it might be possible to do unbiased water splitting with these photoanodes. However in response to these comments, and those of reviewer 2 above, we have carried out unbiased experiments. We have been able to demonstrate that our TiO₂|CsPbBr₃|m-c|GS|WOC devices can generate meaningful photocurrents from water with zero external bias applied (two electrode systems measured over 7 minutes), but unfortunately the levels of O₂ produced were below the detection limit of our oxygen probe. Please see the response to reviewer 2 above for more details.

We have included the new data in the SI (Figure S27) and discussed them in the revised manuscript (Figure 9).

We of course have some ideas for improvements that may bring about said milestone by increasing the photocurrent and allowing us to measure oxygen evolution in the future, but we don't believe that publication of this proof-of-concept communication should depend on its successful demonstration. The focus of the paper remains on the effective, facile and modular protection of a moisture-sensitive perovskite for application in water splitting, and we have used a bias potential to demonstrate record performance and device characteristics that will be of interest to both the PV and water splitting communities.

7. Can the authors split the box plots in Fig S16a in photovoltaic devices used for the photoanodes with and without the WOC? Even if the perovskite batches may have been different (which is not clear at the moment), the average water oxidation photocurrents are equal. Does that point towards kinetic water oxidation limitations?

The average water oxidation photocurrents are equal but the average photovoltaic currents (J_{sc}) of TiO₂|CsPbBr₃|m-c|GS|WOC devices were lower than the ones measured for TiO₂|CsPbBr₃|m-c|GS. As suggested, we have now split the box plots in photovoltaic devices used for photoanodes without the catalyst (CsPbBr₃|m-c|GS) and the ones used for photoanodes with the catalyst (CsPbBr₃|m-c|GS|WOC) in Figure R20, making it clear that perovskite batches were different. **We have also updated the box plots in the revised SI (Figure S25).** We agree with the reviewer's view that the addition of the WOC affords kinetic acceleration of the interfacial charge transfer to the electrolyte

(i.e. e^- extraction from water by surface holes) that shifts the onset potential cathodically, but does not alter the thermodynamics of the device (as shown by identical open circuit potentials). This acceleration still hits a kinetic limit, which may be the maximum turnover frequency of the catalyst, or some other charge transfer barrier across the device.

Figure R20. Box plots of main photovoltaic parameters of $\text{TiO}_2/\text{CsPbBr}_3/\text{m-c}$ later used as photoanodes once GS and GS|WOC protection layers have been applied on the m-c surface.

8. Could the authors add some arrows on Fig. S6b to indicate the three electrodes? The setup is not so clear from the current angle.

We have added arrows to clearly indicate the electrodes in Figure S6b.

9. Fig. S7: It is confusing that the authors zoom-in on the chopped photocurrent in frame c, but they average the light points in frame b. Maybe just show the raw data in frame b as well.

We have changed Figure S7 as suggested.

10. Why do the authors not show the entire chronoamperometric trace from Fig. S12a in Fig. 4a, instead of only arbitrarily showing a part of it?

We intended to draw attention to the initial performance and mid-term stability more than to the drop in photocurrent at the point of failure (which was clearly shown in the SI), but we have now

changed Figure 4 and Figure S13 in the SI to show the entire chronoamperometric trace in the manuscript.

11. Many experiments are run at different pH values (3.5, 6.8, 9, 12.5, then 3.1, 4.3, 7, 10.6 and 12.9 in Fig. S17), with different electrolyte salts (K borate, KNO₃, KOH, some unnamed). While this indicates work conducted by several collaborators independently from each other, could the authors try to give a reason for not using a fixed buffer / buffer range? Can the authors accurately describe all buffers used for the respective experiments?

All experiments were conducted in the same lab by the same researchers, and the variation in electrolytes rather reflects a change in procedures over the course of the study. Initially we used borate buffer solutions as used by Reisner et al. [12], but later found this to be unnecessary so decided to use simple nitrate electrolyte instead. Thus, some of the earlier results in the supplementary information still feature borate buffer although the same results may be obtained without. The electrolyte used throughout all experiments shown in the main manuscript was 0.1 M KNO₃ with the pH adjusted with H₂SO₄ and KOH. The pH of the solution was tested with a calibrated and temperature-compensating pH meter every time before use, which is why values shown vary slightly from experiment to experiment. We have now carefully reviewed all figure captions to ensure that they accurately describe the pH of the solution used in each experiment.

12. How is the open circuit potential (OCP) defined? Is it the potential observed vs Ag/AgCl?

Open circuit potentials (OCP) were measured in a three-electrode system (vs Ag/AgCl reference electrode) by setting the current that flows through the system equal to zero (open circuit condition). The value obtained was then converted to the RHE potential scale using the measured pH of the solution via the Nernst equation. Figure R21 below shows an example of the raw data.

Figure R21. Open Circuit Voltage measurement in a three electrode system with $I=0$ in the dark and under illumination. N.B. the figure shows the drop in open circuit voltage as the light is switched off.

To double check the values obtained, we have also measured the OCP in a two-electrode system. As shown below in Figure R22, we have obtained identical values to the three-electrode measurement.

Figure R22. Comparison of the open circuit voltage measurement of $\text{TiO}_2/\text{CsPbBr}_3/m\text{-c}/\text{GS}$ performed in a three electrode cell (where the potential is measured versus the reference electrode) and in a two electrode cell (where the potential is measured versus the counter electrode).

13. Conclusions: I think the word “oxidation” is missing from “above 70% for direct light-driven water, showing”.

This has now been corrected.

- [1] S. W. Sheehan, J. M. Thomsen, U. Hintermair, R. H. Crabtree, G. W. Brudvig, and C. A. Schmuttenmaer, "A molecular catalyst for water oxidation that binds to metal oxide surfaces.," *Nat. Commun.*, vol. 6, p. 6469, 2015.
- [2] J. M. Thomsen, S. W. Sheehan, S. M. Hashmi, J. Campos, U. Hintermair, R. H. Crabtree, and G. W. Brudvig, "Electrochemical activation of Cp* iridium complexes for electrode-driven water-oxidation catalysis," *J. Am. Chem. Soc.*, vol. 136, no. 39, pp. 13826–13834, 2014.
- [3] Y. Shi, C. Gimbert-Surinach, T. Han, S. Berardi, Ma. Lanza, and A. Llobet, "CuO-Functionalized Silicon Photoanodes for Photoelectrochemical Water Splitting Devices," *Appl. Mater. Interfaces*, vol. 8, pp. 695–702, 2016.
- [4] S. N. Habisreutinger, T. Leijtens, G. E. Eperon, S. D. Stranks, R. J. Nicholas, and H. J. Snaith, "Carbon nanotube/polymer composites as a highly stable hole collection layer in perovskite solar cells," *Nano Lett.*, vol. 14, no. 10, pp. 5561–5568, 2014.
- [5] M. T. Hoang, N. D. Pham, J. H. Han, J. M. Gardner, and I. Oh, "Integrated Photoelectrolysis of Water Implemented on Organic Metal Halide Perovskite Photoelectrode," *ACS Appl. Mater. Interfaces*, vol. 8, no. 19, pp. 11904–11909, 2016.
- [6] J. R. McKone, S. C. Marinescu, B. S. Brunschwig, J. R. Winkler, and H. B. Gray, "Earth-abundant hydrogen evolution electrocatalysts," *Chem. Sci.*, vol. 5, no. 3, pp. 865–878, 2014.
- [7] J. Jin, K. Walczak, M. R. Singh, C. Karp, N. S. Lewis, and C. Xiang, "An experimental and modeling/simulation-based evaluation of the efficiency and operational performance characteristics of an integrated, membrane-free, neutral pH solar-driven water-splitting system," *Energy Environ. Sci.*, vol. 7, no. 10, pp. 3371–3380, 2014.
- [8] Gurudayal, P. M. Chee, P. P. Boix, H. Ge, F. Yanan, J. Barber, and L. H. Wong, "Core – Shell Hematite Nanorods: A Simple Method To Improve the Charge Transfer in the Photoanode for Photoelectrochemical Water Splitting," *ACS Appl. Mater. Interfaces*, vol. 7, no. 12, pp. 6852–6859, 2015.
- [9] Y. Bu, J. Tian, Z. Chen, Q. Zhang, W. Li, and F. Tian, "Optimization of the Photo-Electrochemical Performance of Mo-Doped BiVO₄ Photoanode by Controlling the Metal – Oxygen Bond State on (020) Facet," *Adv. Mater. Interfaces*, vol. 4, p. 1601235, 2017.
- [10] P. Kumar, P. Devi, R. Jain, S. M. Shivaprasad, R. K. Sinha, G. Zhou, and R. Nötzel, "Quantum

dot activated indium gallium nitride on silicon as photoanode for solar hydrogen generation,” *Commun. Chem.*, vol. 2, no. 4, 2019.

- [11] J. W. Moir, E. V. Sackville, U. Hintermair, and G. A. Ozin, “Kinetics versus Charge Separation: Improving the Activity of Stoichiometric and Non-Stoichiometric Hematite Photoanodes Using a Molecular Iridium Water Oxidation Catalyst,” *J. Phys. Chem. C*, vol. 120, no. 24, pp. 12999–13012, 2016.
- [12] M. Crespo-Quesada, L. M. Pazos-Outón, J. Warnan, M. F. Kuehnel, R. H. Friend, and E. Reisner, “Metal-encapsulated organolead halide perovskite photocathode for solar-driven hydrogen evolution in water,” *Nat. Commun.*, vol. 7, p. 12555, 2016.

REVIEWERS' COMMENTS:

Reviewer #1 (Remarks to the Author):

In the revised manuscript, the authors have addressed to the questions raised in the initial reviewer report. Given the inorganic CsPbBr₃-based photoanodes protected by commercial thermal graphite sheet and a mesoporous carbon layer that lead to a record stability in aqueous electrolyte, the manuscript can be considered for publication in Nature Communication in the current form.

Reviewer #2 (Remarks to the Author):

In the first review of the submitted manuscript, this reviewer said that, whereas the paper reported a comprehensive study of the CsPbBr₃-based photoanode protected by carbonaceous materials, this paper did not demonstrate novelty and breakthrough that are required to be publishable in a high-impact journal like Nature Communications, with a few comments. Authors responded and this reviewer carefully reviewed the author's response.

Regarding originality and novelty of the paper, this reviewer still stands by the original evaluation. Even though the paper claims use of some novel materials (CsPbBr₃, mesoporous carbon, graphite sheets, etc), those points seem to result in no major breakthrough.

1) Authors claim that now their photoelectrode can operate for as long as 35 hours, which is somewhat longer than previous reports in the field. (~6 hours) In actual application of the solar fuel technology, however, we're talking about 10-20 years of operation in harsh environments. So, to this reviewer, this paper constitute a step forward, but not a breakthrough, in terms of device reliability and stability.

2) As pointed out in the first review, the motivation of using the high-band gap CsPbBr₃ seems to come from unbiased water splitting. (Otherwise, if normal bandgap material was used instead, much higher current would have occurred.) However, with the current device configuration, the photovoltage is much lower than expected and unbiased photoelectrolysis has been barely achieved even with a catalyst. The solar-to-hydrogen efficiency, one of the most important performance metrics in solar fuel, has not been reported, probably because it is too low (~0.1% estimated from Fig. R12) In conclusion, this reviewer does not recommend the publication of this paper in Nature Communications, as this paper can be published in a more specialized journals.

Reviewer #3 (Remarks to the Author):

My concerns have been addressed and I recommend publication.

Reviewer #1:

In the revised manuscript, the authors have addressed to the questions raised in the initial reviewer report. Given the inorganic CsPbBr₃-based photoanodes protected by commercial thermal graphite sheet and a mesoporous carbon layer that lead to a record stability in aqueous electrolyte, the manuscript can be considered for publication in Nature Communication in the current form.

We thank the reviewer for the positive evaluation of our manuscript, acknowledging the originality and significance of our results.

Reviewer #2:

1. Authors claim that now their photoelectrode can operate for as long as 35 hours, which is somewhat longer than previous reports in the field (~6 hours). In actual application of the solar fuel technology, however, we're talking about 10-20 years of operation in harsh environments. So, to this reviewer, this paper constitute a step forward, but not a breakthrough, in terms of device reliability and stability.

As acknowledged by the reviewer, we demonstrate that our perovskite-based photoelectrode can operate for as long as 35 hours. The longest lifetime reported for a halide perovskite photocathode encapsulated with expensive Field's metal is 20 hours.¹ The longest lifetime reported for a lead halide perovskite photoanode immersed in water like ours is 6 hours, but again with expensive Field's metal. Our metal-free encapsulation design uses inexpensive and commercially available graphite sheets to protect the perovskite-based photoanode, achieving a lifetime which is 15 hours longer than previous reports. In line with the assessments of reviewers 1 and 3, we consider this to be a significant improvement which will trigger lot of research and interest in the field. In addition, we also investigated the reasons for device failure after 35 hours of continuous operation. Engineering aspects such as enhanced graphite sheet resistance and improved sealing techniques may easily lead to much higher stabilities in the near future (work which is currently under way in our laboratories). Whether this will ultimately lead to the 10-20 years operational lifetime as required for real-world application remains to be seen, but as

to date there are no examples of any photoelectrochemical system that come close to meeting these requirements we consider our results to be a significant advance.

Photoelectrochemical water splitting applications aside, we believe that our approach may also be used for fabricating moisture-resistant perovskite solar cells. The limited lifetime of perovskites has long been the major hurdle for wide-spread application of these promising materials in solar cell technology, so using inexpensive, flexible and versatile graphite sheets to enhance the lifetime of perovskite solar cells has the potential to mark a breakthrough in this area too. We have now explicitly added this point to the final discussion in the manuscript (page 21):

The GS protection strategy used here for oxygen evolution in water can also be used to enhance the lifetime of perovskite solar cells, where the flexible nature of graphite may allow for roll-to-roll processing for efficient large scale manufacturing.

2. As pointed out in the first review, the motivation of using the high-band gap CsPbBr₃ seems to come from unbiased water splitting. (Otherwise, if normal bandgap material was used instead, much higher current would have occurred.) However, with the current device configuration, the photovoltage is much lower than expected and unbiased photoelectrolysis has been barely achieved even with a catalyst.

The photovoltage we measured is not 'much lower than expected'. As shown in Figure 7 (page 17), the photovoltage measured for our GS-protected CsPbBr₃ photoanodes was about 1.3 V, very close to its maximum 1.45 V obtained in solar cell configuration. This is in fact a high photovoltage device compared to literature that reports values from 0.54 to 0.86 V.²⁻⁴ As explained in our previous response, the challenge with using this for water splitting is that even the high value of 1.3 V is very close to the thermodynamic threshold of 1.23 V, meaning that less than 0.1 V overpotential is available for driving the kinetics. Contemporary electrolyzers operate at kinetic overpotentials of more than 0.3 V, so it is actually impressive and highly encouraging that we were able to observe any photocurrents under unbiased conditions from a 1.3 V photoanode with a non-photoactive Pt counter electrode. We believe that this will be much appreciated by the scientific community. Optimisation strategies to improve these preliminary results include increasing the catalyst

loading and improving the interfacial charge transport properties. In addition to this, there also have been recent reports showing different approaches to push the open circuit voltage of CsPbBr₃-based devices to up to 1.59 V,⁵⁻⁷ providing increased driving force for unbiased water splitting with our device architecture.

The authors would like to emphasize that there are few single materials that can do measurable unbiased solar water splitting in the literature, and often they require UV light irradiation. Moreover, for a practical and realistic application, the scientific community is targeting tandem devices where for example photoanodes are paired to PV cells or to photocathodes. Pairing our photoanodes with photocathodes will definitely increase the total photovoltage, unbiased current densities and solar-to-hydrogen conversion. In this context, the achievements reported in this paper to produce a stable photoanode with halide perovskites and inexpensive protective layers is a clear breakthrough which will trigger much research in this field.

The solar-to-hydrogen efficiency, one of the most important performance metrics in solar fuel, has not been reported, probably because it is too low (~0.1% estimated from Fig. R12). In conclusion, this reviewer does not recommend the publication of this paper in Nature Communications, as this paper can be published in a more specialized journals.

As the reviewer pointed out, the solar-to-hydrogen (STH) efficiency has not been reported because, as we stated in the manuscript, any amount of oxygen produced in unbiased conditions was below the detection limit of our oxygen probe. On the other hand, in the three-electrode measurements the STH efficiency is meaningless because of the bias applied.⁸ However, under biased conditions we achieved record Faradaic efficiencies >80%, large photocurrents above 2 mAcm⁻² at 1.23 V_{RHE} for 30 hours, and 70% IPCE at 500nm with these non-optimised prototype photoanodes. Thus the approach developed represents a highly promising lead in this area with a clear potential for future application. For the emerging field of halide perovskites in photoanodes for PEC oxygen evolution, with few papers in the literature, we believe this is definitely a breakthrough that Nature Comm. readers will appreciate.

Reviewer #3:

My concerns have been addressed and I recommend publication.

We thank the reviewer for the positive evaluation of our manuscript, recommending publication.

References

- (1) Andrei, V.; Hoyer, R. L. Z.; Crespo-quesada, M.; Bajada, M.; Ahmad, S.; Volder, M. De; Friend, R.; Reisner, E. *Adv. Energy Mater.* **2018**, *8*, 1801403.
- (2) Trzeźniewski, B. J.; Smith, W. A. *J. Mater. Chem. A* **2016**, *4* (8), 2919–2926.
- (3) Du, C.; Yang, X.; Mayer, M. T.; Hoyt, H.; Xie, J.; McMahon, G.; Bischofing, G.; Wang, D. *Angew. Chemie - Int. Ed.* **2013**, *52* (48), 12692–12695.
- (4) Du, C.; Zhang, M.; Jang, J. W.; Liu, Y.; Liu, G. Y.; Wang, D. *J. Phys. Chem. C* **2014**, *118* (30), 17054–17059.
- (5) Duan, J.; Hu, T.; Zhao, Y.; He, B.; Tang, Q. *Angew. Chem. Int. Ed.* **2018**, *57* (20), 5746–5749.
- (6) Duan, J.; Zhao, Y.; Yang, X.; Wang, Y.; He, B.; Tang, Q. *Adv. Energy Mater.* **2018**, *8*, 1802346.
- (7) Yuan, H.; Zhao, Y.; Duan, J.; Wang, Y.; Yang, X.; Tang, Q. *J. Mater. Chem. A* **2018**, *6* (47), 24324–24329.
- (8) Dotan, H.; Mathews, N.; Hisatomi, T.; Gratzel, M.; Rothschild, A. *J. Phys. Chem. Lett.* **2014**, *5*, 3330–3334.